# A Semiparametric Bayesian Method for Sufficient Dimension Reduction

**Abstract**

This work proposes a semiparametric Bayesian approach for statistical inference of the central subspace in the problem of sufficient dimension reduction. Unlike conventional Bayesian approaches for sufficient dimension reduction that model the conditional distributions of the response variable given the projected predictive variables, the new approach models their joint distribution instead via a Dirichlet process Gaussian mixture model, leading to a flexible low-dimensional representation and a tractable posterior sampling strategy. Posterior consistency of the proposed approach is established under the framework of Schwartz's theorem. A Monte Carlo strategy based on the Gibbs sampler and geodesic Monte Carlo is developed for efficient posterior sampling. Simulation studies and real data applications show that the proposed approach is competitive with existing Bayesian and frequentist methods while providing posterior uncertainty quantification for the SDR subspace.

**Keywords:** Semiparametric regression, Single index model, Multiple index model, Dirichlet process, Hamiltonian Monte Carlo.

**Mathematics Subject Classification (2020):** 62F15

## 1 Introduction

High-dimensional data analysis usually faces the "curse of dimensionality" (Bellman, 1961). *Sufficient dimension reduction* (SDR), as pioneered by Li (1991) and Cook (1994), is a path-breaking way of dimension reduction for predictor variables without sacrificing much of its predictive information for the response. Formally, let $X = (X_1, \ldots, X_p)^\intercal \in \mathbb{R}^p$ be a vector of $p$ predictive variables, and $Y \in \mathbb{R}$ be the response variable. Li (1991) introduced sliced inverse regression as a foundational SDR method, while Cook (1994, 1998) systematically developed the notion of dimension reduction subspaces and the central subspace. A common forward-regression representation for SDR is the *multiple index model* (MIM):

$$Y = g(\beta_1^\intercal X, \cdots, \beta_d^\intercal X, \varepsilon), \tag{1}$$

where $g$ is an unknown link function, $\varepsilon$ is an independent random error term, and the model parameter $B = (\beta_1, \cdots, \beta_d)$ forms a $p \times d$ orthonormal matrix. The MIM with $d = 1$ is referred to as the *single index model* (SIM). Considering that general MIM defined in (1) is somewhat difficult to handle, researchers often retreat to the following slightly restricted model with additive noise:

$$Y = g(\beta_1^\intercal X, \cdots, \beta_d^\intercal X) + \varepsilon. \tag{2}$$

Alternatively, Cook (1994) suggested that the same goal can be formulated through conditional independence: there exists a $p \times d$ matrix $A = (a_1, \cdots, a_d)$ such that $Y$ is conditionally independent of $X$ given $A^\intercal X$, i.e.,

$$Y \perp\!\!\!\perp X \mid A^\intercal X, \tag{3}$$

where "$\perp\!\!\!\perp$" denotes independence. According to Cook (1994, 1998), $\mathcal{S}(A) = \text{span}(a_1, \cdots, a_d)$ is called an *SDR subspace* of dimension $d$ if $A$ satisfies (3). Since multiple SDR subspaces may exist, the identifiable target is the *central subspace* $\mathcal{S}$, defined as the intersection of all possible SDR subspaces. Under mild conditions, $\mathcal{S}$ exists, is unique, and is itself the minimal SDR subspace. In the rest of the paper, we reserve $B$ for an orthonormal basis of this target central subspace, so inference is for $\mathcal{S} = \text{span}(B)$ rather than for a unique basis matrix.

Although models (1) and (3) are based on different assumptions, they are equivalent under mild conditions (Zeng and Zhu, 2010). In this context, learning from data means estimating the central subspace $\mathcal{S} = \text{span}(B)$; any matrix $BQ$ with orthogonal $Q \in \mathbb{R}^{d \times d}$ represents the same estimand. Throughout this paper, we assume $d$ is known and refer to $\{\beta_1, \cdots, \beta_d\}$ as the *SDR directions* and $B$ as the *SDR matrix*. We define

$$Z(B) \triangleq B^\intercal X = (\beta_1^\intercal X, \cdots, \beta_d^\intercal X) \tag{4}$$

as the *index vector* with $\beta_j^\intercal X$ being the $j$-th *index variable*. Our goal is to estimate the SDR matrix $B$, which lies on the *Stiefel manifold* $\mathcal{B}_{p,d}$ (consisting of all $p \times d$ orthonormal matrices), or equivalently, the SDR subspace $\mathcal{S} = \text{span}(B)$, which is located in the *Grassmann manifold* $\mathcal{G}_{p,d}$ (consisting of all $d$-dimensional linear subspaces in $\mathbb{R}^p$).

Various methods have been developed for the statistical inference of the SDR subspace $\mathcal{S} = \text{span}(B)$ from both frequentist and Bayesian perspectives. Existing frequentist methods can be classified according to the population object used for identifying the dimension-reduction space. The first category consists of *forward-regression* or conditional-mean approaches, which estimate the central mean subspace through features of $E(Y \mid X)$; representative examples include projection pursuit regression (PPR) (Friedman and Stuetzle, 1981), outer product of gradients (OPG) (Xia et al., 2002), and minimum average variance estimation (MAVE) (Xia et al., 2002, Xia, 2007). The second category consists of *inverse-regression* approaches, which estimate the SDR subspace from features of the conditional distribution of $X$ given $Y$. Classic examples are sliced inverse regression (SIR) (Li, 1991) and sliced average variance estimation (SAVE) (Cook and Weisberg, 1991). These ideas have been extended through contour regression (Li et al., 2005), $L^2$-regularized SIR (Zhong et al., 2005), directional regression (Li and Wang, 2007), sliced regression (Wang and Xia, 2008), and fused estimators based on minimum discrepancy functions (Cook and Zhang, 2014). Sparse and high-dimensional SDR developments include COP (Zhong et al., 2012), SIRI (Jiang and Liu, 2014), DT-SIR (Lin et al., 2018), and Lasso-SIR (Lin et al., 2019). The third category consists of methods that use distributional or semiparametric information beyond the conditional mean. The semiparametric approach of Ma and Zhu (2012) belongs to this category because it uses estimating equations derived from the full conditional distribution $[Y \mid X]$ and can target the central subspace, rather than only modeling $E(Y \mid X)$. Related gradient-based variants use gradients of conditional-distribution

or classification quantities to estimate dimension-reduction spaces beyond the ordinary central mean subspace; examples include coordinate-covariance and kernel-gradient methods (Mukherjee and Zhou, 2006, Mukherjee and Wu, 2006, Wu et al., 2007, Fukumizu and Leng, 2014).

Bayesian SDR methods are less numerous but provide useful posterior uncertainty quantification. Under the forward-regression framework, Tokdar et al. (2010) model the conditional distribution family $\{F_{Y|Z=z}\}_{z\in\mathbb{R}}$ with a logistic Gaussian process, which is then discretized for computation, while Reich et al. (2011) adopt a Gaussian mixture model in which the conditional distributions share common Gaussian components with $z$-specific weights. Related Bayesian dimension-reduction work includes Bayesian nonparametric affine-subspace learning (Page et al., 2013) and Bayesian inverse regression for small datasets (Cai et al., 2019). These approaches illustrate two common computational tensions in Bayesian SDR: direct conditional-density modeling can require stochastic-process discretization over the index space, while mixture-based conditional models can require additional structure for index-dependent weights. These choices may increase computational cost or constrain the flexibility of the conditional distribution.

To address these limitations, we propose a *semiparametric Bayesian* (SPB) method. The inferential target remains the conditional SDR likelihood, but we represent it through the joint distribution of the index vector $Z$ and the response variable $Y$, denoted by $F_{Z,Y}$, using a Dirichlet process Gaussian mixture (DPGM) model. This joint representation recovers $F_{Y|Z}$ by normalization while keeping the nonparametric modeling problem in the low-dimensional $(Z,Y)$ space. Bayesian inference under the new model is derived, with posterior consistency established under mild conditions. An efficient Monte Carlo strategy based on the Gibbs sampler (Liu, 1994, 2004) and geodesic Monte Carlo (Byrne and Girolami, 2013) is developed for posterior sampling. Simulation studies and real data applications show that the proposed approach performs competitively for SDR, especially in nonlinear and multiple-index settings.

The rest of this paper is organized as follows. Section 2 introduces the proposed semiparametric Bayesian models and its inference procedure. Section 3 and Section 4 establish posterior consistency and describe an efficient Monte Carlo strategy, respectively. Section 5 evaluates the proposed method through simulation studies. Section 6 presents real data applications. Finally, section 7 concludes the paper with a brief discussion.

## 2 Semiparametric Bayesian Model and Its Inference

### 2.1 Reparameterization of the SDR model

Under the classic forward regression framework for SDR with parameterization $(B, F_{Y|Z})$, we have the following joint likelihood for an observed dataset composed of $n$ *independent and identically distributed* (i.i.d.) samples $\mathcal{T}_n = \{(x_i, y_i)\}_{1\le i\le n}$:

$$L_n(B, F_{Y|Z}) = \prod_{i=1}^{n} f_{Y|Z}(y_i \mid z_i(B)) \tag{5}$$

where $f_{Y|Z}$ is the density function of $F_{Y|Z}$ and $z_i(B) = B^\mathsf{T} x_i$. While Tokdar et al. (2010) and Reich et al. (2011) choose to model the conditional distribution $F_{Y|Z}$ directly, we keep the same conditional likelihood as the inferential target but introduce an equivalent joint-distribution

parameterization for modeling and computation.

The basic idea is that $F_{Y|Z}$ can be modeled indirectly via $F_{Z,Y}$, the joint distribution of $Z$ and $Y$. For any fixed $B$, this is simply the usual conditional-density identity

$$f_{Y|Z}(y \mid z(B)) = \frac{f_{Z,Y}(z(B), y)}{f_Z(z(B))} = \frac{f_{Z,Y}(z(B), y)}{\int f_{Z,Y}(z(B), u) \mathrm{d}u}.$$

Thus, (5) and the likelihood below are mathematically equivalent; they differ in the object to which we assign a flexible prior. Direct conditional approaches put a prior on the family $\{F_{Y|Z=z} : z \in \mathbb{R}^d\}$, whereas our approach puts a prior on the low-dimensional joint distribution $F_{Z,Y}$ and obtains $F_{Y|Z}$ by conditioning.

Substituting this identity into (5) yields the alternative likelihood, now parameterized by $(B, F_{Z,Y})$:

$$L_n(B, F_{Z,Y}) \triangleq L_n(B, F_{Y|Z}) = \prod_{i=1}^{n} \frac{f_{Z,Y}(z_i(B), y_i)}{\int f_{Z,Y}(z_i(B), y) \, \mathrm{d}y}, \tag{6}$$

This confirms that the classic SDR likelihood, typically parameterized by $(B, F_{Y|Z})$, can be equivalently reparameterized using $(B, F_{Z,Y})$. The denominator is therefore not omitted from the target likelihood; rather, it is the marginal density of $Z$ implied by the joint mixture and will later be handled through a Metropolis-Hastings correction. The advantage of the joint parameterization is technical rather than algebraic. If $f_{Z,Y}$ is modeled by a Gaussian mixture,

$$f_{Z,Y}(z, y) = \sum_{k=1}^{\infty} W_k \phi_{d+1}\{(z, y) \mid \mu_k, \Sigma_k\},$$

then the induced conditional density is also a mixture,

$$f_{Y|Z}(y \mid z) = \sum_{k=1}^{\infty} \omega_k(z) \, \phi\{y \mid m_k(z), s_k^2\},$$

where

$$\omega_k(z) = \frac{W_k \phi_d(z \mid \mu_k^-, \Sigma_k^-)}{\sum_{\ell=1}^{\infty} W_\ell \phi_d(z \mid \mu_\ell^-, \Sigma_\ell^-)}, \quad m_k(z) = \mu_{k,y} + \Sigma_{k,yz}(\Sigma_k^-)^{-1}(z - \mu_k^-),$$

and $s_k^2 = \Sigma_{k,yy} - \Sigma_{k,yz}(\Sigma_k^-)^{-1}\Sigma_{k,zy}$. Therefore the index-dependent weights, means, and variances of $Y \mid Z = z$ are induced automatically by the joint mixture. A direct nonparametric model for $\{f_{Y|Z}(\cdot \mid z)\}$ must instead specify normalized positive functions of $z$ for the weights and often functions of $z$ for the component parameters; logistic-Gaussian-process or other dependent-mixture constructions then require discretization, basis expansion, or nonconjugate updates. These choices are the source of the additional computational cost or reduced flexibility in other Bayesian methods. The identifiable component of this model is the central subspace $\mathcal{S} = \mathrm{span}(B)$. For any orthogonal matrix $Q \in \mathbb{R}^{d \times d}$, $B$ and $BQ$ represent the same point on the Grassmann manifold, and inference below is accordingly reported through $\mathcal{S}$ or through projection-matrix-based summaries.

## 2.2 Prior specification and posterior distribution

Specification of the prior distribution is critical for Bayesian inference. Here, we assume that $B$ and $F_{Z,Y}$ are mutually independent *a priori* with the following joint prior:

$$\pi_0(B, F_{Z,Y}) = \pi_0(B) \cdot \pi_0(F_{Z,Y}),$$

leading to the following posterior distribution:

$$\pi_n(B, F_{Z,Y}) \propto \pi_0(B) \cdot \pi_0(F_{Z,Y}) \cdot \prod_{i=1}^n \frac{f_{Z,Y}(z_i(B), y_i)}{\int f_{Z,Y}(z_i(B), y) \, \mathrm{d}y}. \tag{7}$$

Since no prior knowledge is available for $B$ typically, it is a natural choice to assign a noninformative prior distribution for $B$ in the Stiefel manifold $\mathcal{B}_{p,d}$, i.e.,

$$\pi_0(B) = \mathrm{Unif}(\mathcal{B}_{p,d}) \propto \mathbb{1}(B \in \mathcal{B}_{p,d}), \tag{8}$$

which induces a uniform prior on the Grassmann manifold $\mathcal{G}_{p,d}$. For the special case of $d = 1$, $\mathrm{Unif}(\mathcal{B}_{p,d})$ degenerates to a uniform distribution on the unit sphere $\mathbb{S}^{p-1}$ in $\mathbb{R}^p$.

For the prior distribution of $F_{Z,Y}$, we adopt the *Dirichlet process mixture* (DPM) approach (Ferguson, 1983, Lo, 1984) to specify

$$\pi_0(F_{Z,Y}) = \mathrm{DPM}(\alpha, H, \mathcal{G}), \tag{9}$$

where $\mathcal{G} = \{G_\theta\}_{\theta \in \Theta}$ is a family of $(d+1)$-dimensional distributions for $(Z, Y)$ indexed by $\theta$ which serve as the mixture components. The hyperparameter $\alpha$ and the base distribution $H$ together defines a Dirichlet process on $\Theta$. According to Sethuraman (1994), the *stick-breaking* property of the Dirichlet process leads to the following representation:

$$F_{Z,Y} = \sum_{k=1}^\infty W_k \cdot G_{\theta_k},$$

where $\{\theta_k\}_{k=1}^\infty$ are i.i.d. samples from the base distribution $H$, and the weights are constructed as $W_k = V_k \cdot \prod_{t=1}^{k-1}(1 - V_t)$ with $\{V_k\}_{k=1}^\infty$ being i.i.d. draws from $\mathrm{Beta}(1, \alpha)$.

In this study, we specify $G_\theta = \mathcal{N}(\mu, \Sigma)$, the $(d+1)$-dimensional Gaussian distribution with $\theta = (\mu, \Sigma)$ as the parameters, and choose the base distribution $H$ for $\theta = (\mu, \Sigma)$ to be the *Normal-Inverse-Wishart* distribution $\mathrm{NIW}(\Lambda_0, \nu_0; \mu_0, \kappa_0)$. This leads to the following *Dirichlet process Gaussian mixture* (DPGM) prior for $F_{Z,Y}$ with density

$$
\begin{aligned}
\pi_0(F_{Z,Y}) &= \mathrm{DPGM}\left(\{V_k; \mu_k, \Sigma_k\}_{k=1}^\infty \mid \Xi\right) \\
&= \prod_{k=1}^\infty \left( \mathrm{Beta}(V_k \mid 1, \alpha) \cdot \mathrm{IW}(\Sigma_k \mid \Lambda_0^{-1}, \nu_0) \cdot \mathcal{N}(\mu_k \mid \mu_0, \Sigma_k/\kappa_0) \right) \\
&\propto \prod_{k=1}^\infty \frac{(1 - V_k)^{\alpha-1}}{|\Sigma_k|^{(\nu_0+d+3)/2}} \cdot \exp\left\{ -\frac{1}{2}tr(\Lambda_0 \Sigma_k^{-1}) - \frac{\kappa_0}{2}(\mu_k - \mu_0)^\mathsf{T}\Sigma_k^{-1}(\mu_k - \mu_0) \right\},
\end{aligned} \tag{10}
$$

where $\Xi = (\alpha; \Lambda_0, \nu_0; \mu_0, \kappa_0)$ are hyperparameters, and the status of parameters $\Psi = \{V_k; \mu_k, \Sigma_k\}_{k=1}^{\infty}$ defines a specific joint distribution $F_{Z,Y}$ generated from the prior distribution. Hereinafter, we do not distinguish between the infinite dimensional parameter $\Psi$ and $F_{Z,Y}$, and refer to the support of the DPGM prior as $\mathcal{F}$.

We thus arrive at the Bayesian hierarchical model summarized below:

$$B \sim \mathrm{Unif}(\mathcal{B}_{p,d}), \quad F_{Z,Y} \sim \mathrm{DPGM}(\Xi), \quad y_i \mid x_i, B, F_{Z,Y} \sim f_{Y|Z}\{\,\cdot \mid z_i(B)\,\}, \; 1 \le i \le n,$$

where the sampling procedure of $F_{Z,Y}$ from $\mathrm{DPGM}(\Xi)$ is constructed as follows:

$$
\begin{aligned}
W_k &= V_k \prod_{t<k}(1 - V_t) \text{ with } V_k \sim \mathrm{Beta}(1,\alpha) \text{ for } k < \infty, \\
\Sigma_k &\sim \mathrm{IW}(\Lambda_0^{-1}, \nu_0), \\
\mu_k \mid \Sigma_k &\sim \mathcal{N}(\mu_0, \Sigma_k/\kappa_0), \\
F_{Z,Y} &= \sum_{k=1}^{\infty} W_k \cdot \mathcal{N}(\mu_k, \Sigma_k).
\end{aligned}
$$

This leads to the following posterior distribution of the model parameter $(B, \Psi)$ given the $n$ i.i.d. samples $\mathcal{T}_n = \{(x_i, y_i)\}_{1 \le i \le n}$ from the SDR model:

$$
\begin{aligned}
&\pi_n\left(B, F_{Z,Y}\right) = \pi_n(B, \Psi) \\
\propto\ & \pi_0(B) \cdot \pi_0(F_{Z,Y}) \cdot L_n(B, F_{Z,Y}) \\
\propto\ & \mathbb{1}\left(B \in \mathcal{B}_{p,d}\right) \cdot \mathrm{DPGM}\left(\{V_k; \mu_k, \Sigma_k\}_{k=1}^{\infty} \mid \Xi\right) \cdot \prod_{i=1}^{n} \frac{f_{Z,Y}\left(z_i(B), y_i\right)}{f_Z\left(z_i(B)\right)} \\
\propto\ & \prod_{k<\infty}(1 - V_k)^{\alpha-1} \cdot \prod_{k<\infty} \frac{\exp\left\{-\dfrac{1}{2}tr(\Lambda_0 \Sigma_k^{-1}) - \dfrac{\kappa_0}{2}(\mu_k - \mu_0)^{\mathsf{T}}\Sigma_k^{-1}(\mu_k - \mu_0)\right\}}{|\Sigma_k|^{(\nu_0+d+3)/2}} \\
& \cdot \prod_{i=1}^{n} \left\{ \frac{\sum_{k=1}^{\infty} W_k \cdot \phi\left(z_i(B), y_i \mid \mu_k, \Sigma_k\right)}{\sum_{k=1}^{\infty} W_k \cdot \phi\left(z_i(B) \mid \mu_k^{-}, \Sigma_k^{-}\right)} \right\} \cdot \mathbb{1}(B \in \mathcal{B}_{p,d}), \quad (11)
\end{aligned}
$$

where $\mu_k^{-}$ is the subvector of $\mu$ composed of its first $d$ elements, and $\Sigma_k^{-}$ is the submatrix of $\Sigma_k$ composed of its $d \times d$ elements in the top-left corner. Hereinafter, we refer to this semiparametric Bayesian approach as SPB.

## 2.3 Statistical inference based on posterior samples

Given a group of posterior samples $\{(B^{(t)}, F_{Z,Y}^{(t)}\}_{1 \le t \le T}$ from the SPB model, where $F_{Z,Y}^{(t)}$ is parameterized by $\Psi^{(t)}$, we can obtain posterior samples of the SDR subspace $\mathcal{S}$ by specifying $\mathcal{S}^{(t)} = \mathrm{span}(B^{(t)})$. Based on $\{\mathcal{S}^{(1)}, \cdots, \mathcal{S}^{(T)}\}$, statistical inference about the unknown SDR subspace $\mathcal{S}$ can then be performed.

For example, a point estimation of $\mathcal{S}$ can be obtained by finding the Fréchet mean (Fréchet, 1948), a.k.a. the barycenter, of the posterior samples on the manifold $\mathcal{G}_{p,d}$ with respect to some

distance metric $\mathbf{d}$ defined on $\mathcal{G}_{p,d}$, which minimizes below:

$$\widehat{\mathcal{S}} = \arg\min_{\mathcal{S} \in \mathcal{G}_{p,d}} \sum_{t=1}^{T} \mathbf{d}(\mathcal{S}, \mathcal{S}^{(t)})^2.$$

Reich et al. (2011) (Theorem 3) showed that this optimization problem has an analytical solution

$$\widehat{\mathcal{S}} = \text{span}(\widehat{B}), \tag{12}$$

where $\widehat{B}$ is the first $d$ eigenvectors of the mean projection matrix $\bar{P} = \frac{1}{T}\sum_{t=1}^{T} B^{(t)}B^{(t)\mathsf{T}}$, when $\mathbf{d}(\cdot,\cdot)$ is the projection Frobenius distance

$$\mathbf{d}_{\text{pF}}(\mathcal{S}_1, \mathcal{S}_2) \triangleq \mathbf{d}_{\text{pF}}(B_1, B_2) = ||B_1 B_1^\mathsf{T} - B_2 B_2^\mathsf{T}||_{\text{F}}, \tag{13}$$

where $B_1$ and $B_2$ are the orthonormal bases of $\mathcal{S}_1$ and $\mathcal{S}_2$, and $||\cdot||_{\text{F}}$ is the matrix Frobenius norm.

Based on $\widehat{\mathcal{S}}$, a credible region of $\mathcal{S}$ with a credible level of $\alpha \in (0,1)$ can be obtained by:

$$\mathcal{R}_\alpha = \left\{ \mathcal{S} : \mathbf{d}(\mathcal{S}, \widehat{\mathcal{S}}) \leq \xi_\alpha \right\}, \tag{14}$$

with $\xi_\alpha$ being the $\alpha$-quantile of the empirical distribution $\left\{ \mathbf{d}\left(\mathcal{S}^{(t)}, \widehat{\mathcal{S}}\right) \right\}_{1 \leq t \leq T}$.

Moreover, the proposed model also supports inference about the conditional distribution of $y$ given $x$, i.e., prediction. For a new data point $x^*$, the density of the posterior predictive distribution can be estimated as a Monte Carlo average:

$$\hat{f}(y \mid x^*) = \frac{1}{T}\sum_{t=1}^{T} \frac{f_{Z,Y}^{(t)}(B^{(t)\mathsf{T}}x^*, y)}{f_Z^{(t)}(B^{(t)\mathsf{T}}x^*)}, \tag{15}$$

where $f_Z^{(t)} = \int f_{Z,Y}^{(t)}(B^{(t)\mathsf{T}}x^*, y)dy$ is the marginal density of the first $d$ coordinates. Based on this posterior predictive distribution, the prediction for the unknown response $y$ can be achieved by its expectation:

$$\hat{y} = \frac{1}{T}\sum_{t=1}^{T} \frac{\int y f_{Z,Y}^{(t)}(B^{(t)\mathsf{T}}x^*, y)dy}{f_Z^{(t)}(B^{(t)\mathsf{T}}x^*)}. \tag{16}$$

## 2.4 Selecting the dimension of the SDR space

It is important to determine the dimension $d$ of the SDR space. In the literature, several test-based and cross-validation-based methods have been proposed for the task under the framework of inverse or forward regressions (see Chapters 9 and 10 in Li (2018) for a comprehensive review). Here, we propose using the *Bayesian information criterion* (BIC) introduced by Schwarz (1978) to determine the dimension.

Let $\mathcal{M}_d$ be the candidate model with dimension $d$. The BIC score of $\mathcal{M}_d$ is defined as

$$\text{BIC}(d) = \log(n) \cdot k_d - 2\log(L_d), \tag{17}$$

where $n$ is the sample size, $k_d = dp - \frac{1}{2}d(d+1)$ is the number of free parameters in the parametric

part of $\mathcal{M}_d$, and $L_d$ is the maximized value of the conditional likelihood function under candidate structural dimension $d$. In practice, $L_d$ can be approximated by

$$\widehat{L}_d = \max_{1 \leq t \leq T} \left( \prod_{i=1}^n \left[ \frac{\sum_k W_k^{(t)} \phi(z_i(B^{(t)}), y_i \mid \mu_k^{(t)}, \Sigma_k^{(t)})}{\sum_k W_k^{(t)} \int_{-\infty}^{\infty} \phi(z_i(B^{(t)}), y \mid \mu_k^{(t)}, \Sigma_k^{(t)}) \mathrm{d}y} \right] \right),$$

where $W_k^{(t)} = V_k^{(t)} \prod_{j<k}(1 - V_j^{(t)})$, and $\{B^{(t)}, V_k^{(t)}, \mu_k^{(t)}, \Sigma_k^{(t)}\}$ are posterior samples obtained from the $t$-th iteration of our MCMC algorithm. Then, $d$ can be determined by minimizing the approximated BIC score according to a pregiven upper bound $d_m$, i.e., letting

$$\widehat{d} = \arg \min_{d \leq d_m} \widehat{\mathrm{BIC}}(d),$$

where $\widehat{\mathrm{BIC}}(d)$ is the approximation of $\mathrm{BIC}(d)$ with $L_d$ replaced by $\hat{L}_d$. A full model-selection consistency proof for this semiparametric nonparametric-mixture criterion is beyond the present paper, but the criterion follows the usual BIC separation logic. Suppose the candidate set $\{1, \ldots, d_m\}$ is fixed and contains the true structural dimension $d_0$. If $d < d_0$, the candidate model omits part of the central subspace, and under the uniqueness condition used in Section 3 its best conditional likelihood is separated from that of the true model by a positive Kullback–Leibler gap; the log-likelihood loss is therefore of order $n$, which dominates the $O(\log n)$ BIC penalty. If $d > d_0$, the additional directions are redundant for the conditional law of $Y \mid B^\mathsf{T} X$; under a regular local likelihood expansion, the overfitted model can gain at most a stochastic $O_p(1)$ likelihood improvement, while the extra Stiefel degrees of freedom add an $O(\log n)$ penalty. Hence, under these standard separation and regularity conditions, the BIC rule is expected to favor $d_0$. Establishing a fully rigorous theorem with the DPGM nuisance prior and the Monte Carlo approximation $\hat{L}_d$ is left for future work. The effectiveness of this strategy is validated in our simulation studies and real data applications.

## 3  Posterior Consistency

Let $\mathbf{M}_0 = (B_0, F_{Z,Y}^0)$ denote the true SDR model, formulated as in Section 2, with true parameters $(B_0, F_{Z,Y}^0)$. Let $\mathcal{S}_0 = \mathrm{span}(B_0)$ be the corresponding true SDR subspace. For any $\delta > 0$, we define a $\delta$-neighborhood of the true subspace $\mathcal{S}_0$ as a set of matrices in $\mathcal{B}_{p,d}$:

$$\mathcal{N}_\delta = \{B \in \mathcal{B}_{p,d} : \ \mathbf{d}_{\mathrm{pF}}(\mathrm{span}(B), \mathcal{S}_0) \leq \delta\}, \tag{18}$$

where $\mathbf{d}_{\mathrm{pF}}$ denotes the projection Frobenius distance as defined in (13). For a candidate model $\mathbf{M}$ with parameters $(B, F_{Z,Y})$, let

$$f_\mathbf{M}(x, y) \triangleq f_X^0(x) \frac{f_{Z,Y}(B^\mathsf{T} x, y)}{\int f_{Z,Y}(B^\mathsf{T} x, y) dy} \tag{19}$$

be its corresponding data-generating density, where $f_X^0$ is the true marginal distribution of $X$, and $f_{Z,Y}$ is the density function of $F_{Z,Y}$. The following theorem demonstrates the desired posterior consistency of the proposed method.

**Theorem 1** *Under some regularity conditions (see Supplementary Section A.1.4 for details), the marginal posterior distribution of B, i.e.,*

$$\Pi\left(\mathcal{N}_\delta \mid \mathcal{T}_n\right) = \int_{\mathcal{N}_\delta \times \mathcal{F}} \pi_n(B, F_{Z,Y}) dB dF_{Z,Y},$$

*enjoys posterior consistency. That is,*

$$\lim_{n\to\infty} \Pi\left(\mathcal{N}_\delta \mid \mathcal{T}_n\right) = 1 \ a.s. \ with \ respect \ to \ f_{\mathbf{M}_0}^\infty \ for \ \forall \ \delta > 0,$$

*where $f_{\mathbf{M}_0}^\infty$ is the infinite product of $f_{\mathbf{M}_0}$.*

This theorem guarantees that, as the sample size increases, our semiparametric Bayesian model will correctly identify the true SDR subspace. The detailed proof is deferred to Supplementary Section A.1. The main idea of our proof follows the theoretical framework of Schwartz's theorem (Schwartz, 1965, Ghosal and van der Vaart, 2017) for establishing posterior consistency in nonparametric Bayesian approaches. However, since our framework is semiparametric, we adapt the original proof to suit our specific setting.

The regularity conditions required to establish Theorem 1 include three aspects: an *existence condition* ensuring that the true SDR model $\mathbf{M}_0 = (B_0, F_{Z,Y}^0)$ exists; a *uniqueness condition* ensuring that incorrect SDR subspaces cannot mimic the true data generating process; and a *dense prior condition* ensuring that the prior distribution $\pi_0(B, F_{Z,Y})$ assigns positive mass to neighborhoods of the true model $\mathbf{M}_0$. All of these are mild conditions commonly used in the theoretical analysis of nonparametric Bayesian statistics.

The theorem above is a consistency result. A full optimal posterior contraction-rate theory for the DPGM-induced semiparametric SDR model is substantially more delicate because the rate depends jointly on the nonparametric mixture approximation of $F_{Z,Y}$ and on the geometry of the Grassmann-valued subspace parameter. Nevertheless, the proof structure suggests the following conservative rate interpretation: if the prior small-ball condition and the exponentially consistent tests used in the proof are strengthened to hold at a sequence $\varepsilon_n \to 0$ with $n\varepsilon_n^2 \to \infty$, then the marginal posterior contracts around the true central subspace in projection Frobenius distance at the corresponding rate $\varepsilon_n$. Deriving an explicit minimax-optimal $\varepsilon_n$ for this semiparametric SDR setting remains an important direction for future theoretical work.

## 4 Monte Carlo Strategies for Posterior Sampling

Although a Gibbs sampler iterating between the conditional distributions $\pi_n\left(B \mid F_{Z,Y}\right)$ and $\pi_n\left(F_{Z,Y} \mid B\right)$ is an ideal approach for posterior sampling, it is nontrivial to implement due to the complicated structure of $\pi_n\left(B, F_{Z,Y}\right)$. This section resolves this challenge.

### 4.1 Modify posterior distribution for computational convenience

For statistical models involving DPGM priors, a classic strategy for efficient posterior sampling is to augment the posterior space with a set of latent variables $\{I_i\}_{i=1}^n$, as suggested by Ishwaran and James (2001), where $I_i$ denotes the component indicator for the $i$-th data point. Applying

this idea to the posterior distribution $\pi_n(B, F_{Z,Y})$ in (11) leads to the following augmented posterior distribution:

$$
\pi_n(B, F_{Z,Y}, \{I_i\}_{i=1}^n)
$$

$$
\propto \quad \prod_{k=1}^\infty V_k^{n_k}(1-V_k)^{n_{[>k]}+\alpha-1} \cdot \prod_{k=1}^\infty \frac{\exp\left[-\frac{1}{2}tr(\Lambda_0 \Sigma_k^{-1}) - \frac{\kappa_0}{2}(\mu_k-\mu_0)^\intercal \Sigma_k^{-1}(\mu_k-\mu_0)\right]}{|\Sigma_k|^{(\nu_0+d+3)/2}}
$$

$$
\cdot \frac{\prod_{i=1}^n \phi\left(z_i(B), y_i \mid \mu_{I_i}, \Sigma_{I_i}\right)}{\prod_{i=1}^n \left\{\sum_{k=1}^\infty W_k \cdot \phi\left(z_i(B) \mid \mu_k^-, \Sigma_k^-\right)\right\}} \cdot \mathbb{1}\left(B \in \mathcal{B}_{p,d}\right), \tag{20}
$$

where $n_k = \sum_{i=1}^n \mathbb{1}_{\{I_i=k\}}$ and $n_{[>k]} = \sum_{t>k} n_t$. It is straightforward to check that the augmented posterior distribution $\pi_n(B, F_{Z,Y}, \{I_i\}_{i=1}^n)$ has $\pi_n(B, F_{Z,Y})$ as its marginal distribution. Thus, posterior samples from $\pi_n(B, \Psi)$ can be obtained by simply discarding the $\{I_i\}^{(t)}$ components from the augmented posterior samples.

However, $\pi_n(B, F_{Z,Y}, \{I_i\}_{i=1}^n)$ remains computationally unfriendly. Define the problematic term in the denominator as:

$$
h(B, \Psi) \triangleq \prod_{i=1}^n f_Z(z_i(B)) = \prod_{i=1}^n \left\{\sum_{k=1}^\infty W_k \cdot \phi\left(z_i(B) \mid \mu_k^-, \Sigma_k^-\right)\right\}. \tag{21}
$$

The infinite sum over Gaussian components makes $h(B, \Psi)$ impossible to evaluate directly. Furthermore, the presence of $h(B, \Psi)$ in (20) means that the full conditional distributions are not of standard forms, posing significant challenges for implementing a Gibbs sampler.

To make computation feasible, we replace the DPGM prior with a truncated version ($\text{DPGM}_K$) that allows a maximum of $K$ mixture components. This is achieved by setting $V_K = 1$ in the stick-breaking construction, which forces $W_k = 0$ for all $k > K$. This yields the truncated $\text{DPGM}_K$ prior:

$$
\text{DPGM}_K\left(\{V_k; \mu_k, \Sigma_k\}_{k=1}^K \mid \Xi\right)
$$

$$
\propto \quad \mathbb{1}(V_K=1) \prod_{k<K}(1-V_k)^{\alpha-1} \cdot \prod_{k\le K} \frac{\exp\left\{-\frac{1}{2}tr(\Lambda_0 \Sigma_k^{-1}) - \frac{\kappa_0}{2}(\mu_k-\mu_0)^\intercal \Sigma_k^{-1}(\mu_k-\mu_0)\right\}}{|\Sigma_k|^{(\nu_0+d+3)/2}}.
$$

According to Ishwaran and James (2001), $\text{DPGM}_K$ approximates DPGM well when $K$ is reasonably large. The theoretical consistency result in Section 3 is stated for the non-truncated DPGM prior. The finite-$K$ model used in computation should therefore be understood as a working approximation to the infinite stick-breaking prior. The approximation error is controlled by the residual stick-breaking mass beyond $K$, which decreases geometrically in expectation under the beta stick-breaking construction. In practice, $K$ is chosen large enough that the posterior uses substantially fewer than $K$ occupied mixture components; the sensitivity study in Table 4 further checks this choice empirically. Replacing DPGM with $\text{DPGM}_K$ in our model, we obtain

the truncated augmented posterior as our "working" target distribution:

$$
\pi_n^K(B, F_{Z,Y}, \{I_i\}_{i=1}^n)
$$

$$
\propto \prod_{k<K} V_k^{n_k}(1-V_k)^{n_{[>k]}+\alpha-1} \cdot \prod_{k\leq K} \frac{\exp\left[-\frac{1}{2}tr(\Lambda_0\Sigma_k^{-1}) - \frac{\kappa_0}{2}(\mu_k-\mu_0)^\intercal\Sigma_k^{-1}(\mu_k-\mu_0)\right]}{|\Sigma_k|^{(\nu_0+d+3)/2}}
$$

$$
\cdot \frac{\prod_{i=1}^n \phi\left(z_i(B), y_i \mid \mu_{I_i}, \Sigma_{I_i}\right)}{\prod_{i=1}^n \sum_{k=1}^K W_k \cdot \phi\left(z_i(B) \mid \mu_k^-, \Sigma_k^-\right)} \cdot \mathbb{1}\left(B \in \mathcal{B}_{p,d}\right) \cdot \mathbb{1}\left(V_K = 1\right), \tag{22}
$$

With this truncation, the problematic term $h(B, \Psi)$ simplifies to a computationally tractable form involving only a finite sum:

$$
h(B, \Psi_K) = \prod_{i=1}^n \left\{ \sum_{k=1}^K W_k \cdot \phi(z_i(B) \mid \mu_k^-, \Sigma_k^-) \right\}. \tag{23}
$$

We distinguish three distributions in the computation. The distribution $\pi_n$ denotes the exact posterior under the infinite DPGM prior, $\pi_n^K$ denotes the finite-truncation working target, and $\tilde{\pi}_n^K$ below is not a posterior target but an auxiliary proposal distribution. The denominator in the conditional likelihood is therefore retained in $\pi_n^K$ and is removed only from the proposal, with the resulting discrepancy corrected by a Metropolis-Hastings acceptance probability.

To avoid computational difficulties caused by $h(B, \Psi_K)$, we replace the standard Gibbs proposals for $\pi_n^K(B, \Psi_K, \{I_i\}_{i=1}^n)$ by alternative proposals based on the approximated distribution with $h(B, \Psi_K)$ removed:

$$
\tilde{\pi}_n^K(B, \Psi_K, \{I_i\}_{i=1}^n) = \pi_n^K(B, \Psi_K, \{I_i\}_{i=1}^n) \cdot h(B, \Psi_K)
$$

$$
\propto \prod_{k<K} V_k^{n_k}(1-V_k)^{n_{[>k]}+\alpha-1} \cdot \prod_{k\leq K} \frac{\exp\left[-\frac{1}{2}tr(\Lambda_0\Sigma_k^{-1}) - \frac{\kappa_0}{2}(\mu_k-\mu_0)^\intercal\Sigma_k^{-1}(\mu_k-\mu_0)\right]}{|\Sigma_k|^{(\nu_0+d+3)/2}}
$$

$$
\cdot \prod_{i=1}^n \phi(z_i(B), y_i \mid \mu_{I_i}, \Sigma_{I_i}) \cdot \mathbb{1}\left(B \in \mathcal{B}_{p,d}\right) \cdot \mathbb{1}\left(V_K = 1\right). \tag{24}
$$

The result below shows that $\tilde{\pi}_n^K(B, \Psi_K, \{I_i\}_{i=1}^n)$ is computationally friendly.

**Theorem 2** *Distribution $\tilde{\pi}_n^K(B, \Psi_K, \{I_i\}_{i=1}^n)$ has the following conditional distributions:*

$$
\tilde{\pi}_n^K(I_i = k \mid \cdot) \quad \propto \quad W_k \cdot \phi\left(z_i(B), y_i \mid \mu_k, \Sigma_k\right), \tag{25}
$$

$$
\tilde{\pi}_n^K(V_k \mid \cdot) \quad \sim \quad \text{Beta}\left(n_k + 1, n_{[>k]} + \alpha\right), \tag{26}
$$

$$
\tilde{\pi}_n^K(\mu_k, \Sigma_k \mid \cdot) \quad \sim \quad \text{NIW}(\Lambda_k^*, \nu_k^*; \mu_k^*, \kappa_k^*), \tag{27}
$$

$$
\tilde{\pi}_n^K(\beta_j \mid \cdot) \quad \sim \quad \mathcal{N}(\tilde{\beta}_j, \tilde{M}_j) \cdot \mathbb{1}\left(||\beta_j|| = 1; \beta_j^\intercal \beta_i = 0, i \neq j\right), \tag{28}
$$

*where*

$$\nu_k^* = \nu_0 + n_k, \ \Lambda_k^* = \Lambda_0 + \sum_{I_i=k} (t_i - \bar{t}_k)(t_i - \bar{t}_k)^\intercal + \frac{\kappa_0 n_k}{\kappa_0 + n_k}(\bar{t}_k - \mu_0)(\bar{t}_k - \mu_0)^\intercal,$$

$$\kappa_k^* = \kappa_0 + n_k, \ \mu_k^* = \frac{\kappa_0}{\kappa_0 + n_k} \cdot \mu_0 + \frac{n_k}{\kappa_0 + n_k} \cdot \bar{t}_k,$$

$$\tilde{M}_j = \left[ \sum_{i=1}^n x_i x_i^\intercal \Sigma_{I_i,jj}^{-1} \right]^{-1},$$

$$\tilde{\beta}_j = (\tilde{M}_j)^{-1} \sum_{i=1}^n x_i \left[ \Sigma_{I_i,jj}^{-1} \mu_{I_i,j} - \Sigma_{I_i,j[-j]}^{-1} (B_{[-j]}^\intercal x_i - \mu_{I_i,[-j]}, y_i - \mu_{I_i,d+1}) \right],$$

*with $\bar{t}_k = \sum_{I_i=k} t_i / n_k$, $t_i = (z_i(B), y_i)^\intercal$, $\Sigma_{I_i,jj}^{-1}$ being the $(j,j)$ element of $\Sigma_{I_i}^{-1}$, $\Sigma_{I_i,j[-j]}$ being the $j$-th row of $\Sigma_{I_i}^{-1}$ with the $j$-th element removed, and $B_{[-j]}$ being the submatrix of $B$ with the $j$-th column removed, $\mu_{I_i,j}$ being the $j$-th element of $\mu_{I_i}$, $\mu_{I_i,[-j]}$ being the subvector of $\mu_{I_i}$ with the $j$-th element of $\mu_{I_i}$ removed.*

Apparently, all conditional distributions of $\tilde{\pi}_n^K(B, \Psi_K, \{I_i\}_{i=1}^n)$ are standard distributions that are easy to sample from, except $\pi_n(\beta_j|\cdot)$ in (28).

When $d = 1$, $\pi_n(\beta_j|\cdot)$ degenerates to a $p$-dimensional Gaussian distribution restricted to the unit sphere in $\mathbb{R}^p$, which is known as the $p$-dimensional *Fisher-Bingham distribution* (Kent, 1982). When $d > 1$, extra constraints $\beta_j^\intercal \beta_i = 0$ for any $i \neq j$ enforce the support of $\pi_n(\beta_j|\cdot)$ to collapse into a lower dimensional sphere in the subspace orthogonal to $\mathcal{S}(B_{[-j]})$, leading to a $(p-d+1)$-dimensional Fisher-Bingham distribution. In the literature, a *Hamiltonian Monte Carlo* (HMC) type algorithm called *Geodesic Monte Carlo* (GMC) has been established by Byrne and Girolami (2013) for efficient sampling from distributions on a sphere, making it convenient to conduct Gibbs sampling for $\tilde{\pi}_n^K$.

## 4.2   A Metropolis-within-Gibbs sampler for posterior sampling

The connection between $\tilde{\pi}_n^K$ and $\pi_n^K$ suggests an efficient Metropolis-within-Gibbs sampling strategy. The core idea is to use the full conditional distributions of the distribution $\tilde{\pi}_n^K$ in (25)-(28) as proposal distributions. These proposals are then accepted or rejected using a Metropolis-Hastings correction step, ensuring that the sampler correctly targets the desired posterior distribution $\pi_n^K$. Algorithm 1 implements this Metropolis-within-Gibbs approach, with the corresponding Metropolis-Hastings acceptance ratios derived in Theorem 3.

**Theorem 3** *Given the current status $(B, \Psi_K, \mathbf{I})$ in the Gibbs sampler for the posterior distribution $\pi_n^K(B, \Psi_K, \mathbf{I})$, the Metropolis-Hastings ratios for accepting the conditional moves based on $\tilde{\pi}_n^K(B, \Psi_K, \mathbf{I})$ in Algorithm 1 are:*

$$r(I_i^*) = 1, \quad r(V_k^*) = \min\left\{1, \frac{h(B, \Psi_K)}{h(B, \Psi_K^*(V_k^*))}\right\},$$

$$r(\mu_k^*, \Sigma_k^*) = \min\left\{1, \frac{h(B, \Psi_K)}{h(B, \Psi_K^*(\mu_k^*, \Sigma_k^*))}\right\}, \quad r(\beta_j^*) = \min\left\{1, \frac{h(B, \Psi_K)}{h(B_j^*, \Psi_K)} \cdot r_{GMC}(\beta_j^*)\right\},$$

where $\Psi_K^*(\cdot)$ is the proposed update of $\Psi_K$ according to $\tilde{\pi}_n^K(B, \Psi_K, \mathbf{I})$, $B_j^*$ is the proposed update of $B$ by substituting $\beta_j$ with $\beta_j^*$, and $r_{GMC}$ is the ratio required in the GMC algorithm due to the discretization of the Hamiltonian dynamics.

---

**Algorithm 1** Metropolis-within-Gibbs algorithm for posterior sampling.

---

1: **Hyperparameters**: $\Xi = (\alpha; \Lambda_0, \nu_0; \mu_0, \kappa_0; K)$ and $T$.

2: **Parameters**: $(B, \Psi, \mathbf{I})$ with $B = (\beta_1, \cdots, \beta_d)$, $\Psi = \{V_k, \mu_k, \Sigma_k\}_{k=1}^K$, $\mathbf{I} = \{I_i\}_{i=1}^n$.

3: **Parameter initialization**: $B = B^{(0)}$, $\Psi = \Psi^{(0)}$ and $\mathbf{I} = \mathbf{I}^{(0)}$.

4: **for** $t = 1, \ldots, T$ **do**

5:     For $i = 1, \cdots, n$, sample $I_i^* \sim \tilde{\pi}_n^K(I_i \mid \cdot)$ as in (25), and decide whether to accept $I_k^*$ as $I_i^{(t)}$ or remain at $I_i^{(t-1)}$ based on MH ratio $r(I_i^*)$;

6:     For $k = 1, \cdots, K$, sample $V_k^* \sim \tilde{\pi}_n^K(V_k \mid \cdot)$ as in (26), and decide whether to accept $V_k^*$ as $V_k^{(t)}$ or remain at $V_K^{(t-1)}$ based on MH ratio $r(V_k^*)$;

7:     For $k = 1, \cdots, K$, sample $(\mu_k^*, \Sigma_k^*) \sim \tilde{\pi}_n^K(\mu_k, \Sigma_k \mid \cdot)$ as in (27), and decide whether to accept $(\mu_k^*, \Sigma_k^*)$ as $(\mu_k^{(t)}, \Sigma_k^{(t)})$ or remain at $(\mu_k^{(t-1)}, \Sigma_k^{(t-1)})$ based on MH ratio $r(\mu_k^*, \Sigma_k^*)$;

8:     For $j = 1, \ldots, d$, sample $\beta_j^* \sim \tilde{\pi}_n^K(\beta_j \mid \cdot)$ by GMC (see Supplementary Section A.2.3), and decide whether to accept $\beta_j^*$ as $\beta_j^{(t)}$ or remain at $\beta_j^{(t-1)}$ based on MH ratio $r(\beta_j^*)$.

9: **end for**

10: **Return**: $\left\{ \left( B^{(t)}, \Psi_K^{(t)}, \mathbf{I}^{(t)} \right) \right\}_{0 \leq t \leq T}$.

---

Each Gibbs sweep of Algorithm 1 has mixture-evaluation cost $O\{nK(d+1)^2 + nKp\}$. The first term comes from evaluating and updating the $K$ Gaussian mixture components in the $(d+1)$-dimensional joint index-response space, and the second term comes from forming the index variables and likelihood terms involving the original $p$-dimensional predictors. Thus, for fixed $d$, this part scales linearly in both $n$ and $K$, and it avoids the Gaussian-process discretization required by logistic-Gaussian-process approaches. The direct dense implementation used in the simulations also constructs Fisher–Bingham/GMC proposals for the columns of $B$; this step involves dense linear algebra of order $O\{d(np^2 + p^3)\}$ per sweep. This additional term is modest for the moderate dimensions considered in the paper ($p \leq 50$), but it is the main computational bottleneck in truly high-dimensional settings. For larger $p$, the shrinkage prior in Figure 2 and preliminary screening can reduce the effective dimension, and developing sparse or low-rank linear-algebra implementations is a natural direction for scaling SPB further.

We do not report a raw wall-clock comparison table because such timings are difficult to interpret fairly across the available implementations. Tokdar's implementation is in C, Reich's implementation is in R, and our implementation is in Python. Moreover, MCMC-based methods require choices of iteration number, burn-in, and thinning, while spLGP additionally requires a discretization grid whose resolution trades speed against approximation fidelity. Frequentist iterative methods such as dMAVE and PPR depend on convergence tolerances and random restarts. Consequently, a method can appear faster simply because it was run with fewer iterations, a

coarser grid, or less stringent convergence settings.

## 4.3 Practical issues

Running Algorithm 1 requires specifying the hyperparameters $\Xi = (\alpha; \Lambda_0, \nu_0; \mu_0, \kappa_0; K)$. In practice, we recommend the following default setting:

$$\Lambda_0 = I_{d+1}, \quad \nu_0 = d+1, \quad \mu_0 = \mathbf{0}, \quad \kappa_0 = 1, \quad K = 30. \tag{29}$$

Because both predictors and responses are standardized before fitting, these Normal–Inverse-Wishart hyperparameters are weakly informative default choices for the low-dimensional $(Z, Y)$ mixture components. The hyperparameters with the most direct impact on model complexity or structural sparsity are the truncation level $K$, the DPGM concentration parameter $\alpha$, and the Laplace shrinkage parameter $\lambda$; we therefore focus the sensitivity analyses in Section 5.5, Section 5.6, and Section 5.7 on these quantities. The specification of hyperparameter $\alpha$ is more involved, as it controls the concentration of the DPM: a larger $\alpha$ encourages more components in the mixture model and thus tends to split the data into more clusters. Here, we recommend a data-driven strategy to set $\alpha$. By assigning a $\text{Gamma}(\eta_1, \eta_2)$ prior for $\alpha$, we treat it as a parameter within the Bayesian hierarchical model, with $\eta_1$ and $\eta_2$ as second level hyperparameters. Given $\{V_k\}_{k=1}^{K-1}$, the conditional distribution of $\alpha$ is:

$$\pi_n^K(\alpha \mid \{V_k\}_{k=1}^{K-1}) \sim \text{Gamma}\left(\eta_1 + K - 1, \eta_2 - \sum_{k=1}^{K-1} \log(1 - V_k)\right), \tag{30}$$

based on which $\alpha$ can be updated alongside the iterations of Algorithm 1. Thus, unlike $K$ and $\lambda$, the concentration parameter $\alpha$ is not fixed in our default implementation; it is learned from the data through this conditional update. The second-level hyperparameters $(\eta_1, \eta_2)$ are set to $(1, 1)$ by default. We examine sensitivity to alternative Gamma hyperpriors in Section 5.7.

Standard techniques for MCMC convergence diagnosis based on trace plot analysis of the log-posterior density and parameters (Brooks et al., 2011) can be utilized to assess the convergence of Algorithm 1. Figure 1 visualizes the sampling procedure of Algorithm 1 for a simulated dataset containing 200 data points from a typical single index model with $p = 10$. In this example, the trace plots in Figure 1b suggest that the burn-in period ends after about 2,000 iterations. One can also more formally use the Gelman-Rubin statistic to make such a decision (Gelman and Rubin, 1992). A wide range of simulation studies confirm that such a strategy works effectively in practice.

## 4.4 Inference and computation under shrinkage prior of $B$

Although the noninformative prior for $B$ on the Stiefel manifold $\mathcal{B}_{p,d}$ enjoys conceptual and computational simplicity, a shrinkage prior for $B$ is often preferred when only some of the predictive variables are essential for predicting $Y$. A natural choice is to adopt the Laplace prior constrained on $\mathcal{B}_{p,d}$, defined as:

$$\pi_0(B) \propto \exp\left(-\lambda \|B\|_1\right) \cdot \mathbb{1}(B \in \mathcal{B}_{p,d}),$$

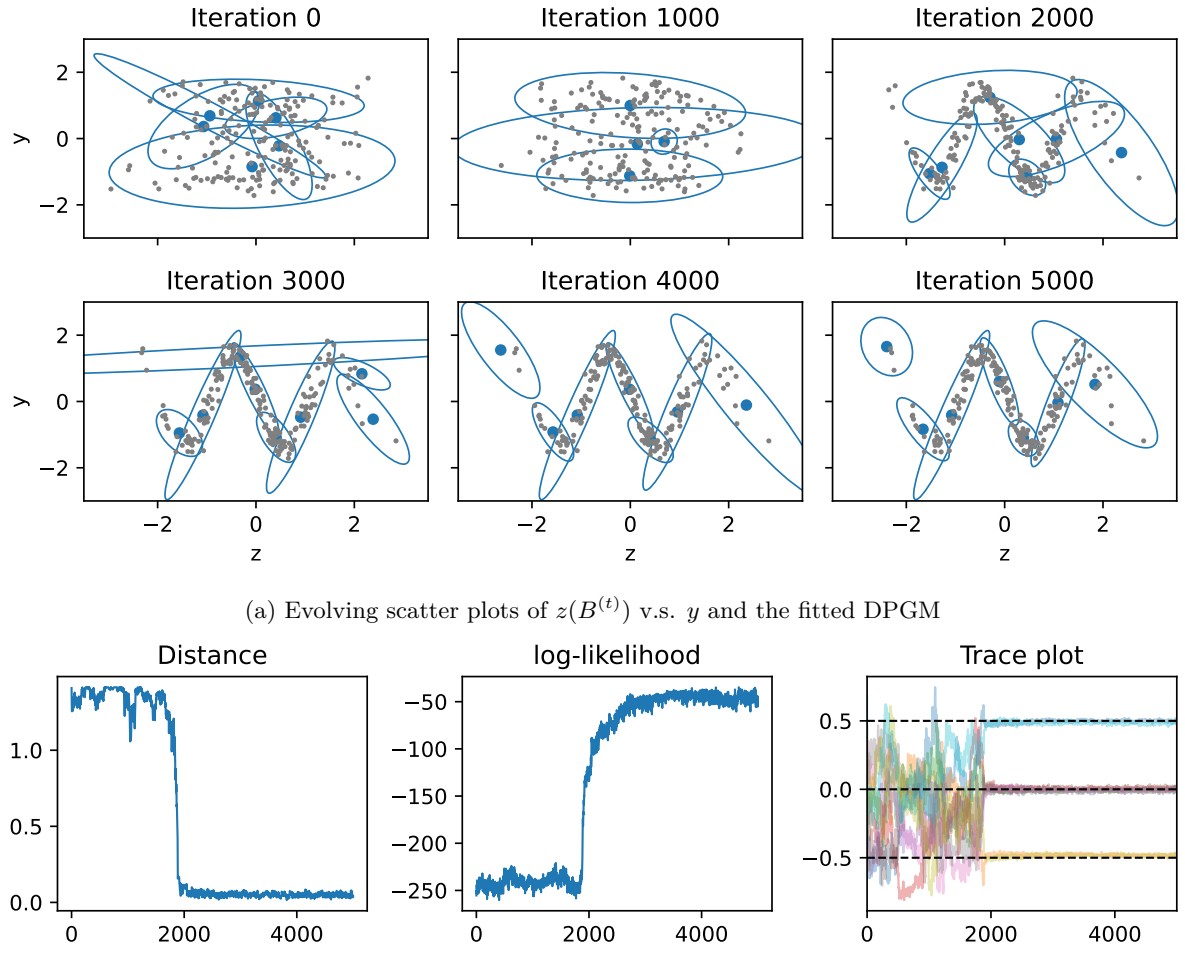

(a) Evolving scatter plots of $z(B^{(t)})$ v.s. $y$ and the fitted DPGM

(b) Distance between $B^{(t)}$ and $B$, log-likelihood, and the trace plot of $B^{(t)}$.

Figure 1: Visualization of the sampling procedure of Algorithm 1 for a simulated dataset with 200 data points from a typical single index model $Y = 2\sin(3B^{\intercal}X) + 0.4\varepsilon$, where $X$ is a 10-dimensional Gaussian random vector, $\varepsilon$ is an independent standard Gaussian random noise and $B = (-.5, .5, 0, 0, 0, 0, 0, 0, .5, -.5)^{\intercal}$.

where $\|B\|_1 = \sum_{i=1}^{p} \sum_{j=1}^{d} |B_{ij}|$ represents the element-wise $L_1$ norm of the matrix $B$, and $\lambda > 0$ controls the strength of shrinkage. Figure 2 provides graphical illustrations of the constrained Laplace prior when $d = 1$ and $p = 3$ with different levels of $\lambda$. As $\lambda$ increases, the prior allocates more probability mass toward the axes (the "vertices") and the great circles of the unit sphere. Clearly, the Laplace prior degenerates to a noninformative uniform prior when $\lambda = 0$.

Bayesian inference of the SDR model under the Laplace prior for $B$ is almost identical to the case under the noninformative prior, except for a slight modification of the GMC step due to the extra term $\exp\left(-\lambda\|B\|_1\right)$, which is discussed in Supplementary Section A.2.3.

In practice, $\lambda = 0$ is appropriate when sparsity in the SDR directions is not expected. When sparse SDR directions are plausible, we recommend selecting $\lambda$ over a modest grid using posterior predictive performance on a validation set or through cross-validation. This criterion is aligned with the predictive interpretation of SDR and avoids treating a single shrinkage value as a universal default.

Simulation studies in section 5 demonstrate that the Laplace prior is an effective shrinkage prior when the hyperparameter $\lambda$ is properly specified.



| $\lambda = 0.1$ | $\lambda = 1$ | $\lambda = 5$ | $\lambda = 10$ |

Figure 2: Visualization of the Laplace prior density on the unit sphere. Bright areas have higher density values than dark areas.

## 4.5 Prediction based on the posterior samples

Given the posterior samples $\{(W_k^{(t)}, \mu_k^{(t)}, \Sigma_k^{(t)})_{k=1}^K, B^{(t)}\}_{t=1}^T$, we can evaluate the model's prediction for a new data point $x^*$. The density of the posterior predictive distribution in (15) can be approximated by the following Monte Carlo estimate:

$$\hat{f}(y|x^*) = \frac{1}{T} \sum_{t=1}^T \frac{\sum_{k=1}^K W_k^{(t)} \cdot \phi\left(B^{(t)\mathsf{T}} x^*, y \mid \mu_k^{(t)}, \Sigma_k^{(t)}\right)}{\sum_{k=1}^K W_k^{(t)} \cdot \phi\left(B^{(t)\mathsf{T}} x^* \mid \mu_k^{(t)^-}, \Sigma_k^{(t)^-}\right)}.$$

Consequently, the prediction for the unknown response $y$ in (16) is estimated as:

$$\hat{y} = \frac{1}{T} \sum_{t=1}^T \frac{\sum_{k=1}^K W_k^{(t)} \cdot \phi\left(B^{(t)\mathsf{T}} x^* \mid \mu_k^{(t)^-}, \Sigma_k^{(t)^-}\right) \cdot \left(\mu_{k,y}^{(t)} + \Sigma_{k,yz}^{(t)} \left(\Sigma_k^{(t)^-}\right)^{-1} \left(B^{(t)\mathsf{T}} x^* - \mu_k^{(t)^-}\right)\right)}{\sum_{k=1}^K W_k^{(t)} \cdot \phi\left(B^{(t)\mathsf{T}} x^* \mid \mu_k^{(t)^-}, \Sigma_k^{(t)^-}\right)},$$

where $\mu_{k,y}^{(t)}$ is the last element of $\mu_k^{(t)}$, and $\Sigma_{k,yz}^{(t)}$ is the last row of $\Sigma_k^{(t)^-}$.

# 5 Simulation Studies

## 5.1 Simulation setting

In this section, we evaluate the performance of the proposed methods via simulation and compare them with existing methods, including BMM (Reich et al., 2011), spLGP (Tokdar et al., 2010), dMAVE (Xia, 2007), PPR (Friedman and Stuetzle, 1981), SIR (Li, 1991), semi-SIR (Ma and Zhu, 2012). The methods considered here are implemented in different software environments: Tokdar's spLGP code is implemented in C, Reich's BMM code is implemented in R, and our prototype implementation is in Python. The codes for BMM and spLGP are obtained from the authors' websites; dMAVE is implemented in the package `MAVE`, PPR is a built-in function in `R`, SIR is in the package `dr`, and its semiparametric version, semi-SIR, is in the package `orthoDr`. In the following simulation studies, all methods are run under their default settings. For iterative methods, the initial value of the SDR space is randomly generated. For all Bayesian methods, including SPB, BMM and spLGP, we conducted 20,000 MCMC iterations for posterior sampling.

**Example 1** *We investigate four single index models $\mathcal{M}_1$-$\mathcal{M}_4$, covering monotone, periodic and symmetric link functions with additive or nonadditive Gaussian noises:*

$$\mathcal{M}_1: \quad Y = \exp(Z/2) + 0.2 \cdot \varepsilon,$$
$$\mathcal{M}_2: \quad Y = 2\sin(2Z) + 0.2 \cdot \varepsilon,$$
$$\mathcal{M}_3: \quad Y = \frac{5}{1 + 2Z^2} + 0.2\left[1 + 2Z^2\right] \cdot \varepsilon,$$
$$\mathcal{M}_4: \quad Y = \sqrt{4 - \min\{Z^2, 4\}} + 0.2 \cdot \varepsilon,$$

*where $Z = \beta^{\mathsf{T}}X$ with the covariates $X \sim N_p(0, I_p)$ and Gaussian noises $\varepsilon \sim N(0, 1)$. The true value of $\beta$ is $\beta = (1/\sqrt{p}, -1/\sqrt{p}, \cdots, (-1)^{p-1}/\sqrt{p})^{\mathsf{T}}$ in $\mathcal{M}_1$, $\beta = (-1/2, 1/2, 0, \cdots, 0, 1/2, -1/2)^{\mathsf{T}}$ in $\mathcal{M}_2$, and $\beta \sim \mathrm{Unif}(\mathbb{S}^{p-1})$ in $\mathcal{M}_3$ and $\mathcal{M}_4$.*

**Example 2** *Next, four multiple index models $\mathcal{M}_5$-$\mathcal{M}_8$ are examined:*

$$\mathcal{M}_5: \quad Y = \frac{1}{0.2 + (Z_1 + 0.5)^2} + \frac{1}{0.2 + (Z_2 - 0.5)^2} + 0.2 \cdot \varepsilon_1,$$
$$\mathcal{M}_6: \quad Y = \mathrm{sign}(2Z_1 + \varepsilon_1)\log(|2Z_2 + 4 + \varepsilon_2|),$$
$$\mathcal{M}_7: \quad Y = Z_1 + 2\sin(Z_2) + 0.2 \cdot \varepsilon_1 + 3 \cdot \mathrm{sign}(\varepsilon_2),$$
$$\mathcal{M}_8: \quad Y = Z_1/2 + \varepsilon_1 \cdot \sqrt{1 - Z_2^2},$$

*where $Z_j = \beta_j^{\mathsf{T}}X$ with the covariates $X \sim N_p(0, I_p)$ and noises $\varepsilon_1, \varepsilon_2 \sim N(0, 1)$. In $\mathcal{M}_5$-$\mathcal{M}_8$, the true SDR vectors are specified to be $\beta_1 = (1/\sqrt{p}, -1/\sqrt{p}, \cdots, (-1)^p/\sqrt{p})^{\mathsf{T}}$, $\beta_2 = (1/2, -1/2, 0, \cdots, 0, 1/2, -1/2)^{\mathsf{T}}$. In $\mathcal{M}_8$, additional constraints apply: $|Z_1| \leq 1$, $|Z_2| \leq 1$, $0.5 < Z_1^2(1 - \varepsilon_1)^2 + \varepsilon_1^2 < 1$. The link functions in $\mathcal{M}_5$-$\mathcal{M}_8$ are visulized in Figure 3.*

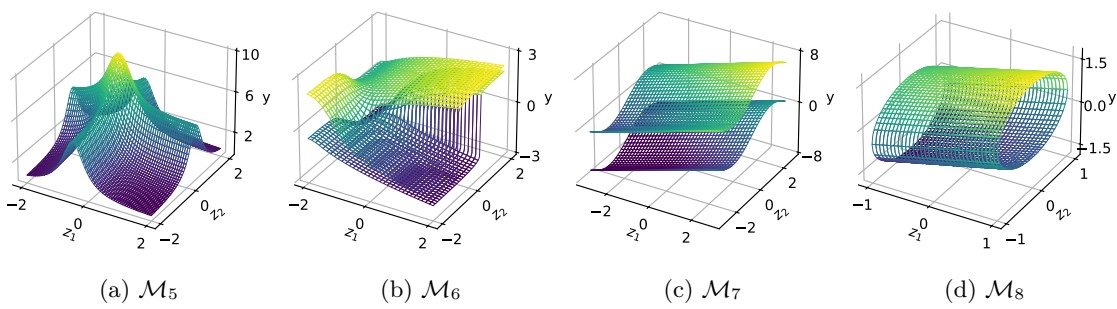

(a) $\mathcal{M}_5$      (b) $\mathcal{M}_6$      (c) $\mathcal{M}_7$      (d) $\mathcal{M}_8$

Figure 3: Link functions of models $\mathcal{M}_5$-$\mathcal{M}_8$. (a) is the mean surface of response $Y$ in model $\mathcal{M}_5$; (b) and (d) is the 10% and 90% quantile surfaces of response $Y$ in model $\mathcal{M}_6$ and $\mathcal{M}_8$; (c) is the surfaces that response $Y$ lies around in model $\mathcal{M}_7$.

## 5.2 SDR subspace estimation

For each of the 8 models, we set $p \in \{10, 20, 50\}$ and the sample size $n \in \{200, 500, 1000\}$, leading to 9 distinct $(p, n)$ simulation settings. For each setting, 100 independent datasets were generated for each model, based on which 7 competing methods were compared. The results are summarized in Table 1 and Table 2, where the values are the means of the projection Frobenius distances between the estimated and the true subspaces in 100 replications. For

each simulation setting (i.e., each row of the table), the mean distances of the top two best methods are highlighted in bold. From these tables, we can see that the proposed method is frequently among the two best methods and remains competitive across a broad range of settings. PPR is highly competitive in some single-index models, especially when the data-generating structure is favorable to projection pursuit, while SPB shows clear advantages in several nonlinear, heteroscedastic, and multiple-index settings. The cases where SPB is not the numerically best method should also be interpreted with some caution. First, in many rows the gap between SPB and the best competitor is very small, so the apparent ordering can be affected by Monte Carlo error, finite-replication randomness, and the stochastic nature of the MCMC approximation. In these cases, the more appropriate conclusion is that the methods are close and comparable rather than that one method has a meaningful advantage. Second, a small fraction of SPB runs may be affected by imperfect MCMC convergence. To keep the large simulation study comparable across all 8 models, 9 $(p, n)$ settings, and 100 replications per setting, we used a fixed MCMC length and common default hyperparameters rather than hand-tuning every individual chain. As a result, some difficult runs may not have mixed fully within the fixed computational budget and can increase the average distance. We have reported these averages without excluding such runs. In a real data analysis with one specific dataset, one would normally inspect trace plots and posterior summaries, tune the MCMC step size or other computational settings if needed, and increase the number of iterations until satisfactory mixing is obtained. Thus, the simulation results should be read as honest fixed-budget average performance, with SPB generally competitive and often among the top methods, rather than as a claim of uniform dominance in every run. Additional simulation results, including convergence diagnostics and posterior inference are reported in Supplementary Section A.3. These results validate the effectiveness of the proposed method for the SDR problem.

## 5.3 Dimension Selection for the SDR subspace

In this subsection, we demonstrate the effectiveness of the proposed BIC criterion in selecting the SDR dimension $d$. Note that models $\mathcal{M}_1$-$\mathcal{M}_4$ are SIMs and models $\mathcal{M}_5$-$\mathcal{M}_8$ are MIMs; thus, the true values of $d$ are 1 and 2, respectively, for them. The parameter $k_d$ in (17) is set to $k_d = dp - \frac{1}{2}d(d+1)$. For the setting with $n = 1,000$ and $p = 50$, we randomly generated 100 replicates of datasets for each model and calculated the BIC values as in (17). The BIC scores for the eight models are displayed in Figure 4. In all the 100 replicates for $\mathcal{M}_1$-$\mathcal{M}_8$, the true value of $d$ is correctly selected.

Table 1: Average projection Frobenius distances ($\times 10$) between the estimated and the true SDR subspaces over 100 repeated experiments in Example 1.

| Setting | | | Methods | | | | | | |
|---------|---|---|------|------|-------|-------|------|------|----------|
| Model | $p$ | $n$ | SPB | BMM | spLGP | dMAVE | PPR | SIR | semi-SIR |
| $\mathcal{M}_1$ | 10 | 200 | **1.05** | 1.52 | 1.39 | 1.38 | **1.03** | 1.41 | 1.92 |
| | | 500 | **0.63** | 1.00 | 0.84 | 0.86 | **0.63** | 0.87 | 1.30 |
| | | 1000 | **0.44** | 0.74 | 0.58 | 0.58 | **0.44** | 0.60 | 0.97 |
| | 20 | 200 | **1.67** | 2.31 | 2.16 | 2.18 | **1.66** | 2.21 | 2.22 |
| | | 500 | **0.95** | 1.42 | 1.23 | 1.25 | **0.92** | 1.28 | 1.55 |
| | | 1000 | **0.64** | 1.04 | 0.87 | 0.89 | **0.65** | 0.87 | 1.19 |
| | 50 | 200 | **3.10** | 3.85 | 4.01 | 8.21 | **3.19** | 4.83 | 3.39 |
| | | 500 | **1.65** | 2.22 | 2.23 | 2.17 | **1.63** | 2.19 | 1.92 |
| | | 1000 | **1.09** | 1.60 | 1.50 | 1.45 | **1.09** | 1.45 | 1.43 |
| $\mathcal{M}_2$ | 10 | 200 | **0.46** | 0.79 | 0.73 | 0.46 | **0.45** | 1.41 | 0.89 |
| | | 500 | **0.25** | 0.50 | 0.48 | 0.26 | **0.25** | 0.81 | 0.81 |
| | | 1000 | **0.17** | 0.36 | 0.32 | 0.18 | **0.17** | 0.60 | 0.81 |
| | 20 | 200 | **0.66** | 0.87 | 0.95 | 0.72 | **0.66** | 2.27 | 0.89 |
| | | 500 | **0.39** | 0.50 | 0.56 | 0.41 | **0.39** | 1.31 | 0.73 |
| | | 1000 | **0.26** | 0.34 | 0.39 | 0.27 | **0.26** | 0.87 | 0.61 |
| | 50 | 200 | 1.25 | **0.95** | 1.56 | **1.09** | 1.22 | 5.04 | 1.29 |
| | | 500 | **0.65** | **0.56** | 0.87 | 0.68 | **0.65** | 2.20 | 0.77 |
| | | 1000 | 0.44 | **0.37** | 0.60 | 0.45 | **0.43** | 1.46 | 0.58 |
| $\mathcal{M}_3$ | 10 | 200 | **0.38** | 1.62 | 1.18 | **0.76** | 7.61 | 13.1 | 1.77 |
| | | 500 | **0.21** | 0.91 | 0.46 | **0.35** | 6.41 | 13.2 | 1.33 |
| | | 1000 | **0.27** | 0.63 | 0.49 | **0.22** | 4.63 | 13.2 | 1.04 |
| | 20 | 200 | **0.62** | 2.35 | 5.52 | **1.19** | 11.1 | 13.8 | 2.32 |
| | | 500 | **0.30** | 1.43 | 1.05 | **0.54** | 7.00 | 13.7 | 1.45 |
| | | 1000 | **0.20** | 1.01 | 0.88 | **0.33** | 6.35 | 13.7 | 1.24 |
| | 50 | 200 | **3.20** | **6.73** | 14.0 | 12.4 | 13.8 | 14.0 | 8.96 |
| | | 500 | **0.56** | 4.02 | 12.2 | **1.00** | 12.6 | 14.0 | 1.93 |
| | | 1000 | **0.49** | 2.57 | 4.84 | **0.56** | 8.31 | 14.0 | 1.34 |
| $\mathcal{M}_4$ | 10 | 200 | **1.26** | 4.81 | 1.62 | 1.88 | 4.07 | 13.1 | **1.28** |
| | | 500 | **0.66** | 8.15 | 0.96 | 1.11 | 2.43 | 13.6 | **0.69** |
| | | 1000 | **0.45** | 7.22 | 0.66 | 0.81 | 1.98 | 13.4 | **0.46** |
| | 20 | 200 | **2.19** | 8.20 | 3.98 | 3.04 | 8.27 | 13.7 | **2.09** |
| | | 500 | **1.41** | 8.34 | 1.82 | 1.61 | 4.57 | 13.7 | **1.10** |
| | | 1000 | **0.67** | 8.18 | 1.08 | 1.11 | 3.00 | 13.7 | **0.71** |
| | 50 | 200 | **5.38** | 12.3 | 13.7 | 12.3 | 13.8 | 14.0 | **6.17** |
| | | 500 | **2.80** | 8.92 | 8.29 | 2.99 | 9.37 | 14.0 | **1.95** |
| | | 1000 | 2.15 | 6.97 | 4.80 | **1.91** | 4.95 | 14.1 | **1.20** |

Table 2: Average projection Frobenius distances ($\times 10$) between the estimated and the true SDR subspaces over 100 repeated experiments in Example 2.

| Setting | | | Methods | | | | | | |
|---|---|---|---|---|---|---|---|---|---|
| Model | $p$ | $n$ | SPB | BMM | spLGP | dMAVE | PPR | SIR | semi-SIR |
| $\mathcal{M}_5$ | 10 | 200 | **1.03** | 3.45 | **1.36** | 1.89 | 10.2 | 14.5 | 3.08 |
| | | 500 | **0.31** | 1.77 | 0.90 | **0.81** | 5.63 | 13.9 | 0.92 |
| | | 1000 | **0.44** | 2.30 | 0.62 | **0.47** | 3.84 | 13.7 | 0.57 |
| | 20 | 200 | **4.30** | 8.03 | 10.5 | **7.43** | 13.7 | 16.2 | 10.5 |
| | | 500 | **0.93** | 2.84 | 2.00 | 1.37 | 10.1 | 14.7 | **1.12** |
| | | 1000 | **0.40** | 2.43 | 1.30 | 0.75 | 5.77 | 14.2 | **0.66** |
| | 50 | 200 | **16.8** | **16.3** | 18.8 | 17.5 | 17.8 | 19.1 | 18.7 |
| | | 500 | **3.66** | 8.46 | 13.6 | **7.27** | 15.1 | 16.7 | 8.99 |
| | | 1000 | 2.23 | 4.38 | 6.99 | **1.32** | 11.7 | 15.3 | **0.71** |
| $\mathcal{M}_6$ | 10 | 200 | 3.95 | 4.80 | **3.93** | **3.71** | 12.5 | 4.58 | 7.58 |
| | | 500 | **2.21** | 3.25 | 2.43 | **2.21** | 9.94 | 2.67 | 5.34 |
| | | 1000 | **1.56** | 3.20 | 1.70 | **1.60** | 7.87 | 1.92 | 3.83 |
| | 20 | 200 | **5.92** | 7.85 | 6.35 | **5.68** | 14.0 | 7.34 | 10.2 |
| | | 500 | **3.50** | 4.73 | 3.89 | **3.53** | 12.4 | 4.22 | 7.15 |
| | | 1000 | **2.39** | 3.55 | 2.68 | **2.46** | 9.91 | 2.87 | 5.69 |
| | 50 | 200 | **10.4** | 11.8 | 11.7 | **11.0** | 16.7 | 15.5 | 16.9 |
| | | 500 | **5.91** | 7.94 | 6.90 | **5.88** | 14.9 | 7.33 | 9.62 |
| | | 1000 | **3.89** | 5.39 | 4.81 | **4.07** | 13.9 | 4.88 | 7.71 |
| $\mathcal{M}_7$ | 10 | 200 | **2.97** | **12.0** | 14.0 | 13.4 | 14.1 | 12.6 | 15.6 |
| | | 500 | **1.11** | **8.28** | 12.2 | 12.8 | 13.6 | 10.8 | 15.0 |
| | | 1000 | **0.66** | **7.25** | 11.2 | 12.1 | 13.0 | 8.33 | 15.9 |
| | 20 | 200 | **7.27** | **13.2** | 16.0 | 14.3 | 15.7 | 14.0 | 16.7 |
| | | 500 | **1.76** | **10.9** | 13.0 | 13.8 | 14.6 | 12.8 | 15.8 |
| | | 1000 | **1.02** | **9.10** | 11.9 | 13.6 | 14.1 | 10.6 | 15.1 |
| | 50 | 200 | **13.4** | **14.4** | 18.7 | 16.8 | 17.4 | 16.3 | 17.7 |
| | | 500 | **3.77** | **13.4** | 14.2 | 14.8 | 16.1 | 14.4 | 16.9 |
| | | 1000 | **1.84** | **11.8** | 12.7 | 14.1 | 15.2 | 13.7 | 15.6 |
| $\mathcal{M}_8$ | 10 | 200 | **3.70** | 8.71 | 10.8 | **5.93** | 13.6 | 13.1 | 16.9 |
| | | 500 | **0.92** | 4.58 | 8.37 | **1.89** | 13.2 | 12.4 | 16.7 |
| | | 1000 | **0.61** | 2.49 | 8.22 | **1.19** | 13.5 | 12.6 | 16.5 |
| | 20 | 200 | **11.6** | 15.7 | 16.0 | 14.8 | **14.4** | **14.4** | 16.8 |
| | | 500 | **3.65** | 11.5 | 13.0 | **3.94** | 14.1 | 13.5 | 16.6 |
| | | 1000 | **3.19** | 5.61 | 12.0 | **1.93** | 13.9 | 13.5 | 17.1 |
| | 50 | 200 | **16.6** | 18.8 | 19.2 | 18.4 | **16.0** | 18.0 | 18.8 |
| | | 500 | **13.6** | 15.9 | 17.9 | 15.6 | **14.9** | 15.0 | 16.8 |
| | | 1000 | **8.27** | 13.4 | 15.1 | **10.9** | 14.4 | 14.2 | 15.5 |

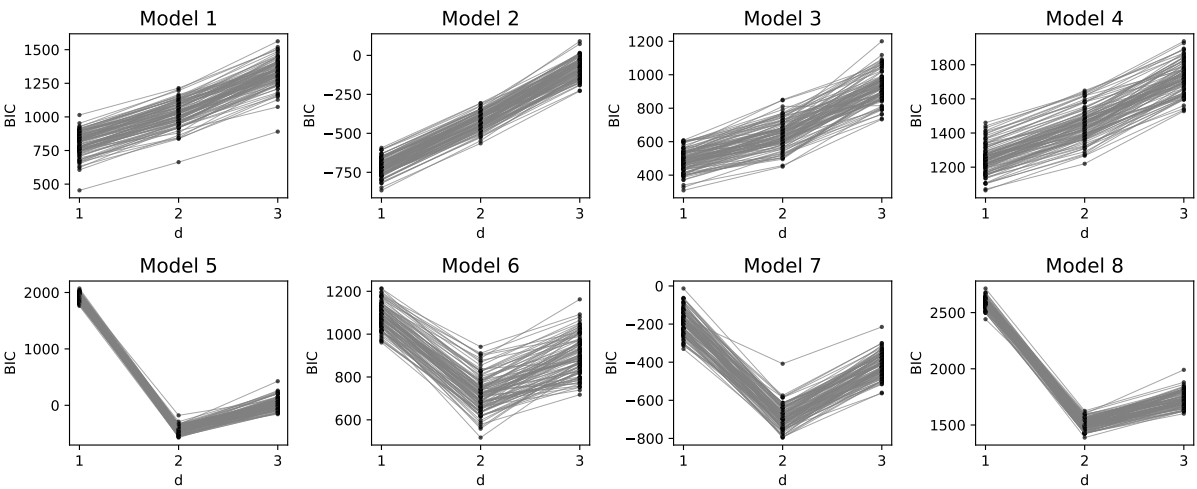

Figure 4: BIC values of different SDR dimensions for the 8 models in Examples 1 and 2.

## 5.4 Shrinkage properties of the Laplace prior

To demonstrate the shrinkage property of the Laplace prior, we consider two illustrative examples:

- A single index model $\mathcal{M}_1$ with $p = 10, d = 1$, and a sparse $\beta$ in which only two of its 10 elements are nonzero, i.e., $\beta = (\frac{\sqrt{2}}{2}, 0, \cdots, 0, \frac{\sqrt{2}}{2})^{\mathsf{T}}$.

- A multiple index model $\mathcal{M}_5$ with $p = 5, d = 2$, and a sparse $B = (\beta_1, \beta_2)$ where only the first two covariates are relevant, i.e., $\beta_1 = (1, 0, 0, 0, 0)^{\mathsf{T}}$ and $\beta_2 = (0, 1, 0, 0, 0)^{\mathsf{T}}$.

Two typical datasets of sample size $n = 100$ are generated from the two models. We then fitted the SPB model using the Laplace-DPGM prior, setting the shrinkage parameter $\lambda$ to two different values: $\lambda = 0$ (which degenerates to the noninformative uniform prior) and $\lambda = 20$. The value $\lambda = 20$ is used here as an illustrative shrinkage setting rather than as a universal default. Comparing the results from these two settings provides useful intuition on the practical impact of the Laplace prior.

Figure 5 compares the marginal posterior distributions of elements in $B$ under the uniform prior ($\lambda = 0$) and the Laplace prior ($\lambda = 20$). The boxplots clearly illustrate the efficacy of the Laplace prior on shrinking irrelevant coefficients toward zero. As shown in Figure 5a, for the 8 irrelevant variables ($\beta_2, \cdots, \beta_9$) in the single index model, the Laplace prior causes their marginal posterior distributions to concentrate sharply at zero compared to those under the uniform prior. Conversely, for the two relevant variables ($\beta_1, \beta_{10}$), the Laplace prior leads to more accurate estimates, as the interference from the irrelevant coefficients has been suppressed by the shrinkage prior. Similarly, Figure 5b (b) shows that for the 3 irrelevant variables in the multiple index model, the posterior distributions of their coefficients shrink markedly toward 0 under the Laplace prior, in contrast to the wide posteriors under the uniform prior. These results highlight the desirable property of the Laplace-DPGM prior: it effectively identifies and preserves the true signals while suppressing the noise from irrelevant variables.

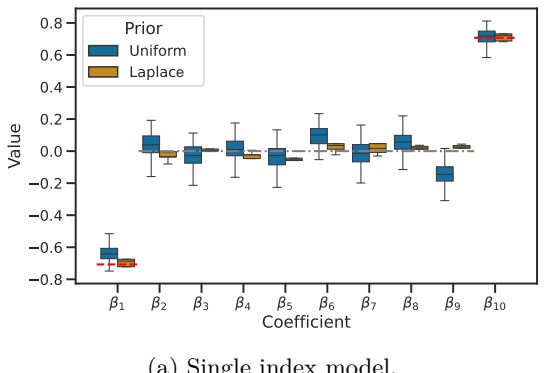

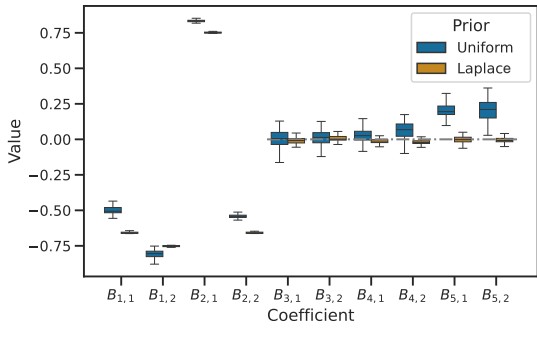

(a) Single index model.  (b) Multiple index model.

Figure 5: Comparison of marginal posterior distributions of elements in $B$ under the uniform prior ($\lambda = 0$) and the Laplace prior ($\lambda = 20$). The gray dashed lines highlight the irrelevant variates, and the red dashed lines highlight the true coefficients of the relevant covariates.

## 5.5 Selection of the shrinkage parameter $\lambda$

We now illustrate the cross-validation procedure for choosing the shrinkage parameter $\lambda$ described in Section 4. We consider a sparse single-index model with an oscillatory link, $Y = 2\sin(2\beta^\mathsf{T}X) + 0.2\varepsilon$, where $X \sim \mathcal{N}(0, I_{10})$ and only two of the ten coefficients are nonzero ($\beta = (\sqrt{2}/2, 0, \ldots, 0, \sqrt{2}/2)^\mathsf{T}$); for such a link the accuracy of the estimated index direction strongly affects prediction, so $\lambda$ is consequential. For each candidate $\lambda$ on a grid, we perform 5-fold cross-validation on a dataset of size $n = 100$ and record the cross-validated negative log predictive density (CV-NLPD) based on the posterior predictive density (15); for reference we also report the cross-validated root mean squared prediction error (CV-RMSE) and the projection Frobenius distance $\mathbf{d}_{\mathrm{pF}}(\widehat{\mathcal{S}}, \mathcal{S}_0)$ to the true subspace, the latter being unavailable in practice.

The results are shown in Table 3. The cross-validated NLPD is clearly U-shaped: it is minimized at $\widehat{\lambda} = 5$ (with the one-standard-error rule giving $\widehat{\lambda} = 10$), the same range that minimizes the projection Frobenius distance to the true subspace, whereas $\lambda = 0$ under-regularizes and $\lambda \geq 20$ over-shrinks and collapses the estimate. The selected $\lambda$ thus simultaneously yields the best prediction and the most accurate subspace recovery, confirming that cross-validation is an effective and fully data-driven way to choose $\lambda$.

Table 3: Selection of the shrinkage parameter $\lambda$ by 5-fold cross-validation on a sparse single-index model with an oscillatory link ($n = 100$, $p = 10$, two relevant predictors). Reported are the cross-validated root mean squared prediction error (CV-RMSE), the cross-validated negative log predictive density (CV-NLPD; standard error in parentheses), and, for reference only, the projection Frobenius distance $\mathbf{d}_{\mathrm{pF}}(\widehat{\mathcal{S}}, \mathcal{S}_0)$ to the true subspace. The smallest CV-NLPD is in bold.

| $\lambda$ | 0 | 5 | 10 | 15 | 20 | 25 | 30 |
|---|---|---|---|---|---|---|---|
| CV-RMSE | 0.501 | 0.272 | 0.282 | 0.519 | 0.679 | 1.205 | 1.387 |
| CV-NLPD | 0.252 | −0.049 | −0.038 | 0.257 | 0.537 | 1.178 | 1.367 |
| (s.e.) | (0.29) | (0.02) | (0.02) | (0.28) | (0.34) | (0.28) | (0.06) |
| $\mathbf{d}_{\mathrm{pF}}(\widehat{\mathcal{S}}, \mathcal{S}_0)$ | 0.043 | 0.036 | 0.046 | 0.053 | 1.408 | 1.401 | 1.399 |

## 5.6 Sensitivity to the truncation level $K$

The DPGM prior is truncated at $K$ components for computation (Section 4.1). We now examine empirically how the posterior of the SDR subspace depends on $K$, complementing the truncation-error bound discussed there. We consider two single-index models with $p = 10$ and $n = 200$: the smooth monotone model $\mathcal{M}_1$, whose joint density $F_{Z,Y}$ is simple, and the non-additive heteroscedastic model $\mathcal{M}_3$, whose joint density is more complex and therefore a more demanding test of the truncation. For each $K \in \{2, 3, 5, 10, 20, 30\}$ we generate 30 datasets, fit the SPB model under the noninformative prior, and record the projection Frobenius distance $\mathbf{d}_{\mathrm{pF}}(\widehat{\mathcal{S}}, \mathcal{S}_0)$ to the true subspace, as well as the number of *occupied* mixture components (those with at least one assigned observation), averaged over the post-burn-in path. We report the median distance over the 30 replications, which is robust to the occasional non-converged chain.

The results are shown in Table 4. Two patterns emerge. First, the posterior of $\mathcal{S}$ is insensitive to $K$ once $K$ is large enough to accommodate the data-generating density: for the simple model $\mathcal{M}_1$ the estimation accuracy is essentially constant across the entire range of $K$, while for the more complex model $\mathcal{M}_3$ the accuracy improves sharply as $K$ increases from 2 to 5 and then stabilizes, with no meaningful change beyond $K = 5$. Second, the number of occupied components saturates well below the truncation level: it levels off at about 4 for $\mathcal{M}_1$ and about 8 for $\mathcal{M}_3$, far short of $K = 20$ or $K = 30$. The unused components confirm that the truncation is not binding at the default value $K = 30$, and that the truncation error is negligible in practice. Together with the exponential bound of Ishwaran and James (2001) cited in Section 4.1, these results justify the use of a moderately large fixed $K$.

Table 4: Sensitivity of the SPB posterior to the truncation level $K$ ($p = 10$, $n = 200$, 30 replications). Reported are the median projection Frobenius distance $\mathbf{d}_{\mathrm{pF}}(\widehat{\mathcal{S}}, \mathcal{S}_0)$ ($\times 10$) to the true subspace and the mean number of occupied mixture components. $\mathcal{M}_1$ is a smooth monotone single-index model and $\mathcal{M}_3$ is a non-additive heteroscedastic one.

| Model | Quantity | $K = 2$ | $K = 3$ | $K = 5$ | $K = 10$ | $K = 20$ | $K = 30$ |
|---|---|---|---|---|---|---|---|
| $\mathcal{M}_1$ | median $\mathbf{d}_{\mathrm{pF}}$ ($\times 10$) | 1.13 | 1.22 | 1.09 | 1.08 | 1.09 | 1.06 |
| | #occupied comp. | 2.0 | 3.0 | 4.2 | 4.1 | 3.9 | 4.2 |
| $\mathcal{M}_3$ | median $\mathbf{d}_{\mathrm{pF}}$ ($\times 10$) | 0.87 | 0.45 | 0.31 | 0.41 | 0.36 | 0.35 |
| | #occupied comp. | 2.0 | 3.0 | 5.0 | 7.9 | 8.2 | 8.1 |

## 5.7 Sensitivity to the DPGM concentration hyperprior

We also examine the robustness of SPB to the concentration parameter $\alpha$ in the DPGM prior. As discussed in Section 4, our default implementation does not fix $\alpha$; instead, it assigns the hyperprior $\alpha \sim \mathrm{Gamma}(\eta_1, \eta_2)$ and updates $\alpha$ within the MCMC sampler. We compare four Gamma hyperpriors, $(\eta_1, \eta_2) \in \{(0.5, 0.5), (1, 1), (2, 1), (5, 1)\}$, using the rate parameterization. The experiment uses the same two dense models as the $K$ sensitivity study, fixes $K = 30$ and $\lambda = 0$, and reports the projection Frobenius distance, validation RMSE, the average number of occupied mixture components, and the posterior mean of $\alpha$.

The results in Table 5 show that the estimated SDR subspace and prediction accuracy are stable across these hyperprior choices. Although the hyperpriors place different prior mass on

Table 5: Sensitivity of SPB to the Gamma hyperprior $\alpha \sim \text{Gamma}(\eta_1, \eta_2)$ for the DPGM concentration parameter, using the rate parameterization. Values are means over 5 simulation replicates, with standard deviations in parentheses. The truncation level is fixed at $K = 30$ and $\lambda = 0$.

| Model | $(\eta_1, \eta_2)$ | $d_{pF}(\widehat{B}, B_0)$ | Validation RMSE | Occupied comp. | Post. mean $\alpha$ |
|---|---|---|---|---|---|
| Dense SIM | $(0.5, 0.5)$ | 0.199 (0.040) | 0.262 (0.062) | 3.600 (0.758) | 1.034 (0.095) |
| Dense SIM | $(1, 1)$ | 0.168 (0.028) | 0.242 (0.039) | 3.485 (0.680) | 1.062 (0.056) |
| Dense SIM | $(2, 1)$ | 0.202 (0.050) | 0.280 (0.058) | 4.805 (0.898) | 0.985 (0.091) |
| Dense SIM | $(5, 1)$ | 0.191 (0.030) | 0.295 (0.100) | 3.616 (1.440) | 1.123 (0.124) |
| Dense MIM | $(0.5, 0.5)$ | 0.264 (0.142) | 1.249 (0.236) | 10.354 (1.917) | 0.913 (0.092) |
| Dense MIM | $(1, 1)$ | 0.189 (0.058) | 1.168 (0.229) | 10.959 (1.519) | 0.837 (0.064) |
| Dense MIM | $(2, 1)$ | 0.250 (0.056) | 1.189 (0.108) | 9.225 (2.266) | 0.907 (0.083) |
| Dense MIM | $(5, 1)$ | 0.166 (0.024) | 0.971 (0.220) | 10.305 (1.983) | 0.961 (0.052) |

the concentration parameter, the posterior mean of $\alpha$ remains in a narrow range, and the number of occupied mixture components changes only moderately. This suggests that, once $K$ is large enough not to bind, the posterior inference for $\mathcal{S}$ is driven primarily by the data rather than by the second-level prior on $\alpha$. The default Gamma$(1, 1)$ setting therefore appears adequate for the simulation regimes considered here.

## 6 Real Data Applications

### 6.1 Auto MPG data

We first apply our method to the Auto MPG dataset, available from https://archive.ics. uci.edu/dataset/9/auto+mpg. This dataset's response variable is city-cycle fuel consumption in miles per gallon (MPG). It includes 7 predictor variables: displacement (DP), cylinders (CL), horsepower (HP), weight (WT), acceleration (AC), model year (MY), and origin (OG). The dataset contains 398 instances, 6 of which have missing values.

The BIC criterion from section 2.4 suggested that a single index model ($d = 1$) is appropriate. We applied the proposed SPB method to the dataset after removing the 6 instances with missing values. The estimated index direction (from the Fréchet mean $\widehat{B}$) yields the index variable:

$$\hat{Z} = -0.012\text{DP} + 0.245\text{CL} + 0.052\text{HP} + 0.006\text{WT} + 0.066\text{AC} - 0.689\text{MY} - 0.677\text{OG}.$$

Predicting MPG using the posterior predictive mean (as in (16)) yielded an $R^2$ of 0.87.

Figure 6 summarizes the main results. We observe that: (a) the relationship between MPG and the estimated index variable $\hat{Z}$ is roughly monotonic, and the predicted mean values appear reasonable (Figure 6a); (b) the 95% credible intervals for the coefficients of model year (MY) and origin (OG) do not contain zero (Figure 6b), indicating that these two covariates are significant. This aligns with the common understanding that newer cars and cars from different regions typically have different fuel efficiencies. Figure 6b also shows that the posterior intervals for horsepower (HP) and weight (WT) miss zero, with the WT interval lying close to zero but still separated from it. Therefore HP and WT should also be regarded as contributory variables for MPG in this fitted single-index model. Their effects are smaller on the standardized index scale than those of MY and OG, and the sign of the displayed coefficients should be interpreted only

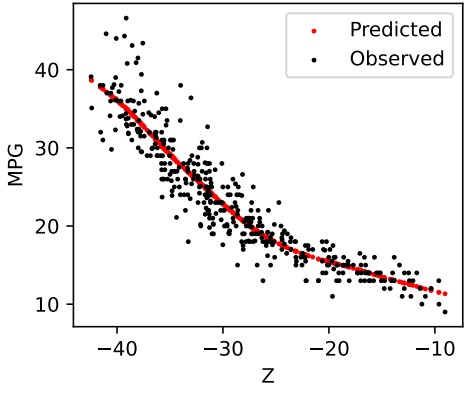

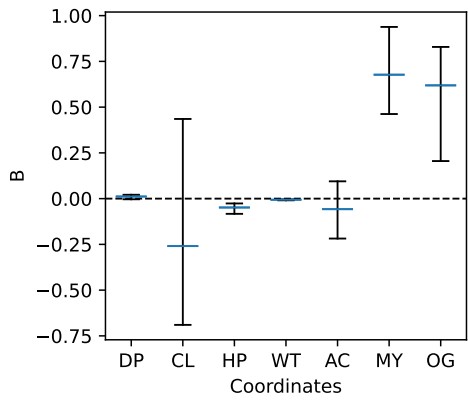

(a) Predicted response curve

(b) Posterior distributions of $B$

Figure 6: Predicted response curve and posterior distributions of index coefficients.

after fixing an arbitrary orientation for the one-dimensional subspace, since $B$ and $-B$ represent the same SDR subspace.

## 6.2 Concrete compressive strength data

Our second real data application uses the concrete compressive strength data reported in Yeh (1998), which contains 1,030 instances. The dataset's response variable is concrete compressive strength (CCS) in MPa, a primary indicator of concrete quality and structural suitability. The 8 predictor variables are cement (CM), blast furnace slag (BFS), fly ash (FA), water (WT), superplasticizer (SP), coarse aggregate (CA), fine aggregate (FAg), and age (AG). The original dataset can be downloaded from: http://archive.ics.uci.edu/ml/datasets/concrete+compressive+strength.

The BIC criterion in Section 2.4 suggested modeling the dataset with a multiple index model where $d = 2$. We applied the proposed SPB method to this dataset. Predicting the response variable CCS with the posterior predictive mean (defined in (16)) yielded $R^2 = 0.848$, as shown in Figure 7a.

Additionally, Figure 7b presents the posterior distribution for each component in the SDR matrix $B$, indicating that all covariates play a role in determining the response. In contrast, predictions based on alternative methods, such as linear regression, PPR, and MAVE, yielded $R^2$ values of 0.616, 0.731, and 0.751, respectively, which are smaller than the 0.848 produced by SPB. Moreover, to address potential over-fitting, we also conducted a 10-fold cross-validation analysis. The results showed that SPB achieved an average Root Mean Square Error (RMSE) of 7.57, which compares favorably with the RMSEs of other models: 10.45 for linear regression, 8.45 for PPR, and 9.20 for dMAVE. These results provide additional evidence that the proposed SPB method can yield accurate predictions in this application. In Figure 7a, the red points are the observed CCS values projected onto the estimated two-index plane, whereas the colored mesh is the posterior predictive mean surface from SPB. The figure is intended to show the nonlinear shape captured by the fitted two-index model; the agreement between observed responses and predictions is summarized quantitatively by the in-sample $R^2$ and the 10-fold cross-validated RMSE reported above.

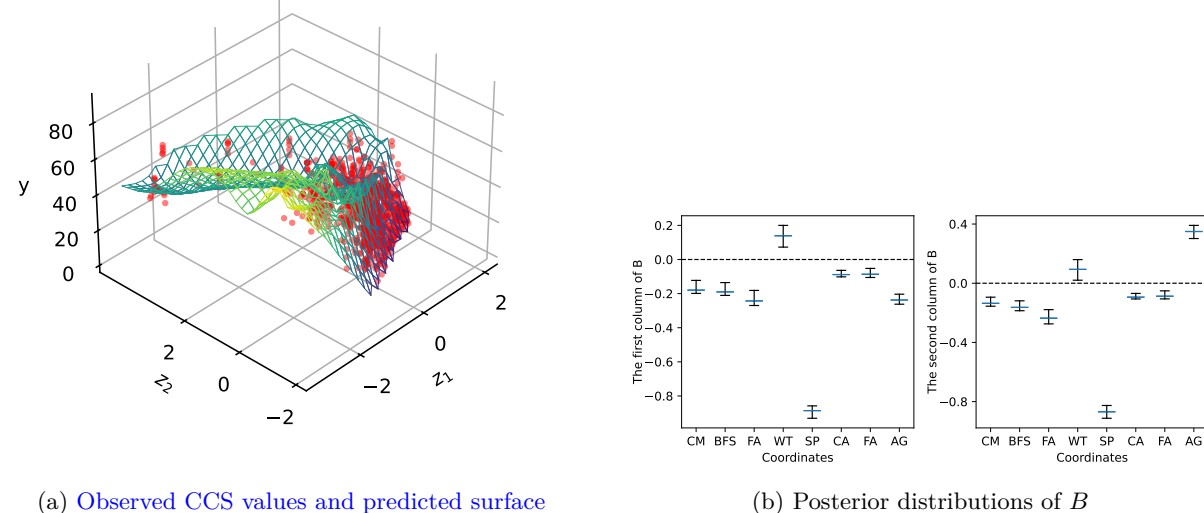

(a) Observed CCS values and predicted surface      (b) Posterior distributions of $B$

Figure 7: Concrete compressive strength analysis. In panel (a), red points are observed responses and the colored mesh is the SPB posterior predictive mean surface. Panel (b) reports posterior distributions of index coefficients.

# 7   Conclusion

In this paper, we proposed a semiparametric Bayesian solution to the sufficient dimension reduction problem. By parameterizing the SDR model through the joint distribution of the index variables and response variable, $F_{Z,Y}$, the proposed method preserves the conditional SDR likelihood while allowing flexible low-dimensional density modeling. Equipping $F_{Z,Y}$ with a Dirichlet process Gaussian mixture prior yields a proper posterior distribution whose consistency can be established under mild regularity conditions. Efficient Monte Carlo strategies for sampling the posterior distribution are developed for both single index models and multiple index models, and posterior samples are used for subspace estimation, uncertainty quantification, dimension selection, shrinkage inference, and prediction. The simulation studies and real data applications show that SPB is competitive with existing Bayesian and frequentist methods, with particular advantages in settings where flexible conditional-distribution fitting is important.

Compared to traditional Bayesian approaches for semiparametric dimension reduction, such as spLGP and BMM, which model the conditional distribution of the response variable given the index variables, the proposed method offers a complementary joint-distribution formulation that leads to tractable posterior computation after finite truncation and Metropolis-Hastings correction. An important limitation is that our current theory establishes consistency but not an explicit optimal posterior contraction rate. In addition, while the BIC rule for selecting $d$ is empirically stable and follows standard likelihood-penalty separation heuristics, a full model-selection consistency theorem with the DPGM nuisance prior remains to be developed. The current implementation is most appropriate for moderate-dimensional problems; scalable sparse linear algebra and screening strategies will be needed for very high-dimensional SDR applications. Deriving contraction rates, formal selection consistency for $d$, and scalable high-dimensional implementations remain useful directions for future work.

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

# A  Supplemental Material

## A.1  Proof of Theorem 1: Posterior Consistency

### A.1.1  Two key concepts

First, we introduce the concept of *KL property*, which plays a critical role in defining a proper prior distribution for nonparametric Bayesian statistics. For any two SDR models $\mathbf{M}$ and $\mathbf{M}'$ with $(B, F_{Z,Y})$ and $(B', F'_{Z,Y})$ as the model parameters respectively, we define their KL divergence as the KL divergence of their data generating distributions defined in (19), i.e.,

$$\text{KL}\left(\mathbf{M}||\mathbf{M}'\right) \triangleq \text{KL}(f_{\mathbf{M}}||f_{\mathbf{M}'}) = -\int f_{\mathbf{M}}(x,y) \log \frac{f_{\mathbf{M}'}(x,y)}{f_{\mathbf{M}}(x,y)} dx dy. \tag{S1}$$

Further, define $\mathcal{M} \subset \mathcal{B}_{p,d} \times \mathcal{F}_{Z,Y}$ as a set of SDR models with different parameters. For two sets of SDR models $\mathcal{M}_1$ and $\mathcal{M}_2$, their KL divergence is defined as the minimum KL divergence of a pair of models from them, i.e.,

$$\text{KL}\left(\mathcal{M}_1||\mathcal{M}_2\right) \triangleq \inf_{\mathbf{M}_1 \in \mathcal{M}_1, \mathbf{M}_2 \in \mathcal{M}_2} \text{KL}\left(\mathbf{M}_1||\mathbf{M}_2\right). \tag{S2}$$

Now, we formally define the *KL property* of a semiparametric Bayesian SDR model:

**Definition 1** *(KL property) A semiparametric Bayesian SDR model $\mathbf{M}$ with $(B, F_{Z,Y})$ as the parameters is said to possess the KL property with respect to the prior distribution $\Pi_0$ if, for any $\varepsilon > 0$, there exists a measurable subset $\mathcal{M}$ in the model space such that $KL(\mathbf{M}||\mathcal{M}) \leq \varepsilon$ and $\Pi_0(\mathcal{M}) > 0$.*

Next, we introduce the *testability condition* that helps explicitly describe the uniqueness of the true SDR model $\mathbf{M}_0$:

**Definition 2** *(Testability condition) For a series of SDR model sets $\{\mathcal{M}_n\}_{1 \leq n < \infty}$, we say that the true model $\mathbf{M}_0 = (B_0, F_{Z,Y}^0)$ is testable against them if, for any collection of $n$ i.i.d. samples $\mathcal{T}_n = \{(x_i, y_i)\}_{1 \leq i \leq n}$ from $\mathbf{M}_0$ and the following hypothesis testing problem*

$$H_0 : \ \mathcal{T}_n \sim f_{\mathbf{M}_0} \quad v.s. \quad H_1 : \ \mathcal{T}_n \sim f_{\mathbf{M}} \text{ for some } \mathbf{M} \in \mathcal{M}_n, \tag{S3}$$

*there exist a constant $C > 0$ and a randomized test function $\phi_n : (\mathbb{R}^{p+1})^n \to [0,1]$ where a small value of $\phi_n$ close to zero suggests rejecting $H_0$, s.t.*

$$\lim_{n \to \infty} \phi_n(\mathcal{T}_n) = 0 \ a.s. \ w.r.t. \ f_{\mathbf{M}_0}^{\infty} \quad and \quad \sup_{\mathbf{M} \in \mathcal{M}_n} \mathbb{E}_{f_{\mathbf{M}}}(1 - \phi_n(\mathcal{T}_n)) \leq e^{-Cn}, \tag{S4}$$

*where the expectation $\mathbb{E}_{f_{\mathbf{M}}}$ is for $\mathcal{T}_n$ with respect to the distribution $f_{\mathbf{M}}$ defined in (19).*

### A.1.2  The first lemma

Connecting the KL property and the testability condition, the lemma below provides a general result to establish posterior consistency for the joint posterior distribution of SPB.

**Lemma 1** *If the true SDR model $\mathbf{M}_0 = (B_0, F_{Z,Y}^0)$ possesses the KL property with respect to the prior distribution $\Pi_0$, then for any series of measurable SDR model sets $\{\mathcal{M}_n\}_{1 \leq n < \infty}$ satisfying the testability condition, their posterior distribution satisfies*

$$\lim_{n \to \infty} \Pi(\mathcal{M}_n \mid \mathcal{T}_n) = 0 \ a.s. \ with \ respect \ to \ f_{\mathbf{M}_0}^\infty.$$

**Proof:** For a dataset $\mathcal{T}_n = \{(x_i, y_i)\}_{1 \leq i \leq n}$ from the semiparametric Bayesian model $\mathbf{M} = (B, F_{Z,Y})$, we represent its likelihood function $L_n(B, F_{Z,Y})$ in the following compact form

$$L_n(\mathbf{M}) = \prod_{i=1}^n f_{Y|Z}\left(y_i \mid B'x_i\right),$$

and define the likelihood ratio between model $\mathbf{M}$ and the true model $\mathbf{M}_0 = (B_0, F_{Z,Y}^0)$ as

$$R_n(\mathbf{M}) = \frac{L_n(\mathbf{M})}{L_n(\mathbf{M}_0)}.$$

For a model set $\mathcal{M}_n$ in a series of measurable model sets $\{\mathcal{M}_n\}_{1 \leq n < \infty}$, its posterior mass is

$$\Pi(\mathcal{M}_n \mid \mathcal{T}_n) = \frac{\int_{\mathcal{M}_n} L_n(\mathbf{M}) d\Pi_0(\mathbf{M})}{\int_{\mathcal{B} \times \mathcal{F}} L_n(\mathbf{M}) d\Pi_0(\mathbf{M})} = \frac{\int_{\mathcal{M}_n} R_n(\mathbf{M}) d\Pi_0(\mathbf{M})}{\int_{\mathcal{B} \times \mathcal{F}} R_n(\mathbf{M}) d\Pi_0(\mathbf{M})} \in [0, 1].$$

Because the corresponding randomized test function $\phi_n(\mathcal{T}_n) \in [0, 1]$ as well, it is straightforward to check that

$$\Pi(\mathcal{M}_n \mid \mathcal{T}_n) \leq \phi_n(\mathcal{T}_n) + (1 - \phi_n(\mathcal{T}_n))\Pi(\mathcal{M}_n \mid \mathcal{T}_n). \tag{S5}$$

Thus, we can prove $\Pi(\mathcal{M}_n \mid \mathcal{T}_n) \to 0$ a.s. by showing that both terms on the right-hand side of the above inequality converge to 0. According to the assumption of the testability condition in Definition 2, the convergence of the first term $\phi_n(\mathcal{T}_n)$ is obvious.

To show the convergence of the second term in (S5), we analyze its numerator and denominator separately. First, we consider the numerator. By Fubini's theorem and the testability condition, there exists a constant $C > 0$ s.t.

$$\mathbb{E}_{f_{\mathbf{M}_0}}\left[(1 - \phi_n(\mathcal{T}_n)) \int_{\mathcal{M}_n} R(\mathbf{M}) d\Pi_0(\mathbf{M})\right] = \int_{\mathcal{M}_n} \mathbb{E}_{f_{\mathbf{M}_0}}\left[(1 - \phi_n(\mathcal{T}_n)) R(\mathbf{M})\right] d\Pi_0(\mathbf{M})$$

$$= \int_{\mathcal{M}_n} \mathbb{E}_{f_{\mathbf{M}}}\left[1 - \phi_n(\mathcal{T}_n)\right] d\Pi_0(\mathbf{M}) \leq e^{-Cn}.$$

Therefore, for any $0 < c' < C$,

$$\sum_{n=1}^\infty e^{c'n} \mathbb{E}_{f_{\mathbf{M}_0}}\left[(1 - \phi_n(\mathcal{T}_n)) \int_{\mathcal{M}_n} R(\mathbf{M}) d\Pi_0(\mathbf{M})\right] \leq \sum_{n=1}^\infty e^{-(C-c')n} < \infty.$$

By Markov's inequality, this implies that for any $\epsilon > 0$ we have

$$\sum_{n=1}^\infty \mathbb{P}\left(e^{c'n}(1 - \phi_n(\mathcal{T}_n)) \int_{\mathcal{M}_n} R(\mathbf{M}) d\Pi_0(\mathbf{M}) > \epsilon\right) < \infty.$$

Then, by the Borel-Cantelli lemma we have that

$$e^{c'n}(1 - \phi_n(\mathcal{T}_n)) \int_{\mathcal{M}_n} R(\mathbf{M})d\Pi_0(\mathbf{M}) \to 0, \quad a.s. \tag{S6}$$

Next, we consider the denominator. According to the KL property of the true model $\mathbf{M}_0$, there exists a constant $c \in (0, C)$ and a model set $\mathcal{M}$ s.t.

$$\mathrm{KL}(\mathbf{M}_0 || \mathcal{M}) \leq c < C \quad \text{and} \quad 0 < \Pi_0(\mathcal{M}) \leq 1.$$

Because $R_n(\mathbf{M}) > 0$, we have

$$\int_{\mathcal{B} \times \mathcal{F}} R_n(\mathbf{M})d\Pi_0(\mathbf{M}) \geq \int_{\mathcal{M}} R_n(\mathbf{M})d\Pi_0(\mathbf{M}) = \Pi_0(\mathcal{M}) \int_{\mathcal{M}} R_n(\mathbf{M})d\Pi_0(\mathbf{M} \mid \mathcal{M}), \tag{S7}$$

where $d\Pi_0(\mathbf{M} \mid \mathcal{M}) = \frac{d\Pi_0(\mathbf{M})}{\Pi_0(\mathcal{M})}$ is the restriction of the prior distribution $\Pi_0$ on $\mathcal{M}$. By Jensen's inequality, Fubini's theorem, and the Strong Law of Large Numbers (SLLN), we have

$$\frac{1}{n} \log \left[ \int_{\mathcal{M}} R_n(\mathbf{M})d\Pi_0(\mathbf{M} \mid \mathcal{M}) \right]$$
$$\geq \frac{1}{n} \int_{\mathcal{M}} \log R_n(\mathbf{M})d\Pi_0(\mathbf{M} \mid \mathcal{M})$$
$$= \frac{1}{n} \sum_{i=1}^{n} \left[ \int_{\mathcal{M}} \log \frac{f_{Y|Z}(y_i \mid B'x_i)}{f_{Y|Z}^0(y_i \mid B_0'x_i)} d\Pi_0(\mathbf{M} \mid \mathcal{M}) \right]$$
$$\to \mathbb{E}_{f_{\mathbf{M}_0}} \left[ \int_{\mathcal{M}} \log \frac{f_{Y|Z}(y_i \mid B'x_i)}{f_{Y|Z}^0(y_i \mid B_0'x_i)} d\Pi_0(\mathbf{M} \mid \mathcal{M}) \right] \quad a.s.$$
$$= -\int_{\mathcal{M}} KL(\mathbf{M}_0 || \mathbf{M})d\Pi_0(\mathbf{M} \mid \mathcal{M}) \geq -c,$$

which implies

$$\lim_{n \to \infty} \left[ e^{cn} \int_{\mathcal{M}} R_n(\mathbf{M})d\Pi_0(\mathbf{M} \mid \mathcal{M}) \right] \geq 1. \tag{S8}$$

Based on (S7) and (S8), we have for any $c' \in (c, C)$ that

$$e^{c'n} \int_{\mathcal{B} \times \mathcal{F}} R_n(\mathbf{M})d\Pi_0(\mathbf{M}) \geq e^{c'n} \int_{\mathcal{M}} R_n(\mathbf{M})d\Pi_0(\mathbf{M} \mid \mathcal{M})$$
$$= e^{(c'-c)n} \left[ e^{cn} \int_{\mathcal{M}} R_n(\mathbf{M})d\Pi_0(\mathbf{M} \mid \mathcal{M}) \right] \to \infty \quad a.s.,$$

which implies

$$e^{c'n} \int_{\mathcal{B} \times \mathcal{F}} R_n(\mathbf{M})d\Pi_0(\mathbf{M}) \to \infty \ a.s.. \tag{S9}$$

Finally, combining equation S6 and equation S9, we get the following limiting behavior of the second term in (S5):

$$(1 - \phi_n(\mathcal{T}_n)) \frac{\int_{\mathcal{M}_n} R_n(\mathbf{M})d\Pi_0(\mathbf{M})}{\int_{\mathcal{B} \times \mathcal{F}} R_n(\mathbf{M})d\Pi_0(\mathbf{M})} = \frac{e^{c'n}(1 - \phi_n(\mathcal{T}_n)) \int_{\mathcal{M}_n} R(\mathbf{M})d\Pi_0(\mathbf{M})}{e^{c'n} \int_{\mathcal{B} \times \mathcal{F}} R_n(\mathbf{M})d\Pi_0(\mathbf{M})} \to 0 \ a.s.,$$

which completes the proof. ∎

### A.1.3    The second lemma

To prove Theorem 1 based on Lemma 1, we only need to show that $\mathcal{N}_\delta^c \times \mathcal{F}$, where $\mathcal{N}_\delta^c$ stands for the complementary set of $\mathcal{N}_\delta$, satisfies the testability condition as a constant series of model sets. The next lemma provides a positive answer to this request.

**Lemma 2** *Suppose that $\psi_j : \mathbb{R}^{p+1} \to [0,1]$, where $1 \le j \le M \in \mathbb{N}$, are a finite number of bounded continuous functions. For any $\varepsilon > 0$, define*

$$\mathcal{U}_\varepsilon \triangleq \left\{ \mathbf{M} \in \mathcal{B} \times \mathcal{F} : |\mathbb{E}_{f_\mathbf{M}} \psi_j - \mathbb{E}_{f_{\mathbf{M}_0}} \psi_j| < \varepsilon, \ j = 1, 2, \dots, M \right\}. \tag{S10}$$

*Then $\mathcal{U}_\varepsilon^c$, as a constant series of model sets, satisfies the testability condition defined in Definition 2.*

**Proof:** According to the definition of the testability condition, we need to construct a test function $\phi_n$ for the hypothesis testing problem in (S3) that satisfies the conditions in (S4).

For the continuous bounded function $\psi_j : \mathbb{R}^{p+1} \to [0,1]$, define

$$\mathcal{U}_{\varepsilon,j}^{(1)} = \left\{ \mathcal{M} \in (\mathcal{B} \times \mathcal{F}) : \mathbb{E}_{f_\mathbf{M}} \psi_j < \mathbb{E}_{f_{\mathbf{M}_0}} \psi_j + \varepsilon \right\},$$
$$\mathcal{U}_{\varepsilon,j}^{(2)} = \left\{ \mathcal{M} \in (\mathcal{B} \times \mathcal{F}) : \mathbb{E}_{f_\mathbf{M}} (1 - \psi_j) < \mathbb{E}_{f_{\mathbf{M}_0}} (1 - \psi_j) + \varepsilon \right\}.$$

It is straightforward to see that the set $\mathcal{U}_\varepsilon$ defined in (S10) can be represented as:

$$\mathcal{U}_\varepsilon = \bigcap_{j=1}^{M} \left( \mathcal{U}_{\varepsilon,j}^{(1)} \bigcap \mathcal{U}_{\varepsilon,j}^{(2)} \right).$$

For $\mathcal{U}_{\varepsilon,j}^{(1)}$ and $\mathcal{U}_{\varepsilon,j}^{(2)}$, we construct the following test functions for the hypothesis testing problem defined in (S3) respectively:

$$\phi_{n,j}^{(1)}(\mathcal{T}_n) = \mathbb{1} \left\{ \frac{1}{n} \sum_{i=1}^{n} \psi_j(x_i, y_i) > \mathbb{E}_{f_{\mathbf{M}_0}} \psi_j + \frac{\varepsilon}{2} \right\},$$
$$\phi_{n,j}^{(2)}(\mathcal{T}_n) = \mathbb{1} \left\{ \frac{1}{n} \sum_{i=1}^{n} (1 - \psi_j(x_i, y_i)) > \mathbb{E}_{f_{\mathbf{M}_0}} (1 - \psi_j) + \frac{\varepsilon}{2} \right\}.$$

Considering that $0 \le \phi_{n,j}^{(\xi)} \le 1$ for $\forall \, j \in \{1, 2, \cdots, M\}$ and $\xi \in \{1, 2\}$, by Hoeffding's inequality and the definition of $\mathcal{U}_{\varepsilon,j}^{(1)}$ and $\mathcal{U}_{\varepsilon,j}^{(2)}$, we have the following exponentially bounded type I and type II error rates for $\phi_{n,j}^{(1)}$ and $\phi_{n,j}^{(2)}$:

$$\mathbb{E}_{f_{\mathbf{M}_0}} \left( \phi_{n,j}^{(1)}(\mathcal{T}_n) \right) \le e^{-n\varepsilon^2/2} \quad \text{and} \quad \mathbb{E}_{f_\mathbf{M}} \left( 1 - \phi_{n,j}^{(1)}(\mathcal{T}_n) \right) \le e^{-n\varepsilon^2/2} \text{ for } \forall \, \mathbf{M} \in \left( \mathcal{U}_{\varepsilon,j}^{(1)} \right)^c,$$
$$\mathbb{E}_{f_{\mathbf{M}_0}} \left( \phi_{n,j}^{(2)}(\mathcal{T}_n) \right) \le e^{-n\varepsilon^2/2} \quad \text{and} \quad \mathbb{E}_{f_\mathbf{M}} \left( 1 - \phi_{n,j}^{(2)}(\mathcal{T}_n) \right) \le e^{-n\varepsilon^2/2} \text{ for } \forall \, \mathbf{M} \in \left( \mathcal{U}_{\varepsilon,j}^{(2)} \right)^c.$$

To verify the testability condition for $\mathcal{U}_\varepsilon^c$, we construct the following test function for the hypothesis testing problem defined in (S3):

$$\phi_n(\mathcal{T}_n) = \max_{1 \le j \le M, \, \xi = 1, 2} \left\{ \phi_{n,j}^{(\xi)}(\mathcal{T}_n) \right\}.$$

By the linearity and monotonicity of expectation, we obtain the following bounds for the type I and type II error rates of $\phi_n$:

$$
\mathbb{E}_{f_{\mathbf{M}_0}}(\phi_n(\mathcal{T}_n)) \leq \sum_{1 \leq j \leq M, \ \xi=1,2} \mathbb{E}_{f_{\mathbf{M}_0}}\left(\phi_{n,j}^{(\xi)}(\mathcal{T}_n)\right) \leq 2Me^{-n\varepsilon^2/2}, \tag{S11}
$$

$$
\mathbb{E}_{f_{\mathbf{M}}}\left(1 - \phi_n(\mathcal{T}_n)\right) \leq \min_{1 \leq j \leq M, \ \xi=1,2} \mathbb{E}_{f_{\mathbf{M}}}\left(1 - \phi_{n,j}^{(\xi)}(\mathcal{T}_n)\right) \leq e^{-n\varepsilon^2/2} \text{ for } \forall \ \mathbf{M} \in \mathcal{U}_{\varepsilon}^c. \tag{S12}
$$

Based on (S11), we have

$$
\sum_{n=1}^{\infty} \mathbb{E}_{f_{\mathbf{M}_0}}(\phi_n(\mathcal{T}_n)) \leq 2M \sum_{n=1}^{\infty} e^{-n\varepsilon^2/2} < \infty.
$$

which implies the first requirement in the testability condition (S4), i.e.,

$$
\lim_{n \to \infty} \phi_n(\mathcal{T}_n) = 0 \text{ a.s. w.r.t. } f_{\mathbf{M}_0}^{\infty},
$$

according to Markov's inequality and the Borel-Cantelli lemma. Moreover, because (S12) implies the second requirement in the testability condition (S4), i.e.,

$$
\sup_{\mathbf{M} \in \mathcal{U}_{\varepsilon}^c} \mathbb{E}_{f_{\mathbf{M}}}\left(1 - \phi_n(\mathcal{T}_n)\right) \leq e^{-n\varepsilon^2/2},
$$

the proof is complete. ∎

### A.1.4 Formal regularity conditions

With the above preparations, we can formally introduce the three regularity conditions required by Theorem 1.

**Condition 1** *(Existence) There exists an SDR model $\mathbf{M}_0$ with parameter $(B_0, F_{Z,Y}^0)$ such that $f_{\mathbf{M}_0}(x,y) = f_X^0(x) \frac{f_{Z,Y}^0(B_0^{\mathsf{T}}x, y)}{\int f_{Z,Y}^0(B_0^{\mathsf{T}}x, y)dy} = f_{X,Y}^0(x,y)$.*

**Condition 2** *(Uniqueness) For any $\varepsilon > 0$, there exists a model set $\mathcal{U}_{\varepsilon}$, as defined in (S10), such that $d_{pF}(span(B), \mathcal{S}_0) > \varepsilon$ implies that $\mathbf{M} = (B, F_{Z,Y}) \in \mathcal{U}_{\varepsilon}^c$ for any $F_{Z,Y} \in \mathcal{F}$.*

**Condition 3** *(Dense prior) $\mathbf{M}_0$ possesses the KL property with respect to the prior distribution $\Pi_0$.*

### A.1.5 Proof of Theorem 1

**Proof:** For any $\delta > 0$ and the corresponding size-$\delta$ neighborhood of $\mathcal{S}_0$ in $\mathcal{G}_{p,d}$, i.e., $\mathcal{N}_{\delta}$, as defined in (18), we have

$$
\mathbf{d}_{\text{pF}}\left(\text{span}(B), \mathcal{S}_0\right) \leq \delta,
$$

according to the definition of $\mathcal{N}_{\delta}$. Based on the Condition 2 (uniqueness), this fact implies that there exists a model set $\mathcal{U}_{\delta}$, as defined in (S10), such that $\mathbf{M} = (B, F_{Z,Y}) \in \mathcal{U}_{\delta}^c$ for any $B \in \mathcal{N}_{\delta}$ and $F_{Z,Y} \in \mathcal{F}$, or equivalently $\mathcal{N}_{\delta}^c \times \mathcal{F} \subseteq \mathcal{U}_{\delta}^c$.

Because Theorem 2 already confirms that $\mathcal{U}_\delta^c$ satisfies the testability condition, $\mathcal{N}_\delta^c \times \mathcal{F}$, as a subset of $\mathcal{U}_\delta^c$, also satisfies the testability condition. Then, as a consequence of Theorem 1, we have:

$$\lim_{n\to\infty} \Pi\left(\mathcal{N}_\delta^c \times \mathcal{F} \mid \mathcal{T}_n\right) = 0 \text{ a.s. w.r.t. } f_{\mathbf{M}_0}^\infty \text{ for } \forall \, \delta > 0.$$

Considering that

$$\Pi\left(\mathcal{N}_\delta \times \mathcal{F} \mid \mathcal{T}_n\right) + \Pi\left(\mathcal{N}_\delta^c \times \mathcal{F} \mid \mathcal{T}_n\right) = \Pi\left(\mathcal{S} \times \mathcal{F} \mid \mathcal{T}_n\right) = 1,$$

we have

$$\Pi\left(\mathcal{N}_\delta \times \mathcal{F} \mid \mathcal{T}_n\right) = 1 - \Pi\left(\mathcal{N}_\delta^c \times \mathcal{F} \mid \mathcal{T}_n\right) \to 1 \text{ a.s. w.r.t. } f_{\mathbf{M}_0}^\infty \, \forall \, \delta > 0,$$

which completes the proof.

## A.2 Technical Details of Monte Carlo Strategies for SPB

### A.2.1 Proof of Theorem 2: posterior calculation

The joint simplified posterior distribution is given by (24). With $B = (\beta_1, \ldots, \beta_d)$ fixed, the conditional distribution $\tilde{\pi}_n^K(\Psi_K, \{I_i\}_{i=1}^n \mid B)$ is a regular $(d+1)$-dimensional truncated DPGM posterior for density estimation based on the data $\{(z_i(B), y)\}_{i=1}^n$, which is essentially a finite Gaussian mixture.

First, with other parameters fixed, $I_i$ only appears in the $i$-th likelihood $W_{I_i} \cdot f_{Z,Y}(z_i(B), y_i \mid I_i)$, as in (22). As a result, the full conditional of $I_i$ is a discrete distribution with probability

$$\tilde{\pi}_n^K\left(I_i = k \mid \cdot\right) \propto W_k \cdot \phi\left(z_i(B), y_i \mid \mu_k, \Sigma_k\right), k = 1, 2, \ldots, K.$$

Second, the full conditional of $V_k$ is also straightforward to derive. $V_k$ only appears in $V_k^{n_k}(1 - V_k)^{n_{[>k]}+\alpha-1}$, which is a Beta distribution with parameters $(n_k + 1, n_{[>k]} + \alpha)$.

Third, $(\mu_k, \Sigma_k)$ appears in both the Normal-Inverse-Wishart prior

$$\frac{\exp\left[-\frac{1}{2}tr(\Lambda_0 \Sigma_k^{-1}) - \frac{\kappa_0}{2}(\mu_k - \mu_0)^\intercal \Sigma_k^{-1}(\mu_k - \mu_0)\right]}{|\Sigma_k|^{(\nu_0+d+3)/2}},$$

and the multivariate Gaussian likelihood

$$\prod_{\{i:I_i=k\}} \phi(z_i(B), y_i \mid \mu_k, \Sigma_k).$$

The full conditional distribution of $(\mu_k, \Sigma_K)$ is again a Normal-Inverse-Wishart distribution because of the conjugate prior. The parameters of this posterior conditional Normal-Inverse-Wishart distribution follow from standard Bayesian analysis of the multivariate Gaussian likelihood with an NIW prior.

The above full conditionals are all either analytically easy to derive or simply conjugate.

However, the conditional distributions of $\beta_j, j = 1, \ldots, d$, require a bit careful calculation:

$$\tilde{\pi}_n^K(\beta_j \mid \Psi_k, \{I_i\}_{i=1}^n, B_{-j})$$

$$\propto \prod_{i=1}^n \phi(z_i(B), y_i \mid \mu_{I_i}, \Sigma_{I_i}) \cdot \mathbb{1}\left(B \in \mathcal{B}_{p,d}\right)$$

$$\propto \exp\left\{-\frac{1}{2}\sum_{i=1}^n [(B^\intercal x_i, y_i) - \mu_{I_i}]^\intercal \Sigma_{I_i}^{-1} [(B^\intercal x_i, y_i) - \mu_{I_i}]\right\} \cdot \mathbb{1}\left(B \in \mathcal{B}_{p,d}\right)$$

$$\propto \prod_{i=1}^n \exp\left\{ -\frac{1}{2}\beta_j^\intercal x_i \Sigma_{I_i,j,j}^{-1} x_i^\intercal \beta_j - \beta_j^\intercal x_i \Sigma_{I_i,j,[-j]}^{-1}\left((B_{[-j]}^\intercal x_i, y_i) - \mu_{I_i,[-j]}\right)\right.$$

$$\left. + \beta_j^\intercal x_i \Sigma_{I_i,j,j}^{-1} \mu_{I_i,j}\right\} \cdot \mathbb{1}\left(\|\beta_j\| = 1, \beta_j \perp \beta_j', j' \neq j\right).$$

The exponential part is quadratic in terms of $\beta_j$. As a result, the conditional distribution of $\beta_j$ is a Gaussian distribution with restrictions. The mean vector and covariance matrix can be derived by completing the quadratic form, as in theorem 2.

### A.2.2 Proof of Theorem 3: Metropolis-Hastings ratios

Let $\tau(\boldsymbol{\theta})$ be the target distribution to be sampled, and $\tilde{\tau}(\boldsymbol{\theta})$ be an approximation of $\tau(\mathbf{x})$ whose conditional distributions are easier to sample from. We propose moves from the current status $\boldsymbol{\theta}$ to a new status $\boldsymbol{\theta}^*$, denoted as $\boldsymbol{\theta} \to \boldsymbol{\theta}^*$, based on the conditional distributions of $\tilde{\tau}(\boldsymbol{\theta})$, i.e., utilizing the proposal distribution

$$g(\boldsymbol{\theta}^* \mid \boldsymbol{\theta}) = \tilde{\tau}(\theta_j^* \mid \boldsymbol{\theta}_{[-j]}) \cdot \mathbb{1}(\boldsymbol{\theta}_{[-j]} = \boldsymbol{\theta}_{[-j]}^*),$$

leading to the Metropolis-Hastings ratio below:

$$r(\boldsymbol{\theta} \to \boldsymbol{\theta}^*) = \min\left\{1, \frac{\tau(\boldsymbol{\theta}^*) \cdot g(\boldsymbol{\theta} \mid \boldsymbol{\theta}^*)}{\tau(\boldsymbol{\theta}) \cdot g(\boldsymbol{\theta}^* \mid \boldsymbol{\theta})}\right\}.$$

Define

$$\rho(\boldsymbol{\theta}) = \frac{\tilde{\tau}(\theta_j \mid \boldsymbol{\theta}_{[-j]})}{\tau(\theta_j \mid \boldsymbol{\theta}_{[-j]})}.$$

We have

$$\frac{\tau(\boldsymbol{\theta}^*) \cdot g(\boldsymbol{\theta} \mid \boldsymbol{\theta}^*)}{\tau(\boldsymbol{\theta}) \cdot g(\boldsymbol{\theta}^* \mid \boldsymbol{\theta})} = \frac{\tau(\theta_j^* \mid \boldsymbol{\theta}_{[-j]}) \cdot \tilde{\tau}(\theta_j \mid \boldsymbol{\theta}_{[-j]})}{\tau(\theta_j \mid \boldsymbol{\theta}_{[-j]}) \cdot \tilde{\tau}(\theta_j^* \mid \boldsymbol{\theta}_{[-j]})} \cdot \mathbb{1}(\boldsymbol{\theta}_{[-j]} = \boldsymbol{\theta}_{[-j]}^*)$$

$$= \frac{\rho(\boldsymbol{\theta})}{\rho(\boldsymbol{\theta}^*)} \cdot \mathbb{1}(\boldsymbol{\theta}_{[-j]} = \boldsymbol{\theta}_{[-j]}^*).$$

In our case, considering that $\boldsymbol{\theta} = (B, \Psi_K, \mathbf{I})$, and

$$\rho(\boldsymbol{\theta}) = \frac{\tilde{\tau}(\theta_j \mid \boldsymbol{\theta}_{[-j]})}{\tau(\theta_j \mid \boldsymbol{\theta}_{[-j]})} = h(B, \Psi_K) = \prod_{i=1}^n \left\{\sum_{k=1}^K W_k \cdot \phi(z_i(B) \mid \mu_k^-, \Sigma_k^-)\right\},$$

we have

$$
\frac{\tau(\boldsymbol{\theta}^*) \cdot g(\boldsymbol{\theta} \mid \boldsymbol{\theta}^*)}{\tau(\boldsymbol{\theta}) \cdot g(\boldsymbol{\theta}^* \mid \boldsymbol{\theta})} = \begin{cases} \frac{h(B, \Psi_K)}{h(B^*, \Psi_K)}, & \text{if } \boldsymbol{\theta}^* \text{ and } \boldsymbol{\theta} \text{ differ at } B\text{-related dimensions only,} \\ \frac{h(B, \Psi_K)}{h(B, \Psi_K^*)}, & \text{if } \boldsymbol{\theta}^* \text{ and } \boldsymbol{\theta} \text{ differ at } \Psi\text{-related dimensions only,} \\ 1, & \text{if } \boldsymbol{\theta}^* \text{ and } \boldsymbol{\theta} \text{ differ at } \mathbf{I}\text{-related dimensions only.} \end{cases}
$$

Because we use the GMC algorithm to sample $B$ instead of directly sampling it like other parameters, another ratio $r_{\text{GMC}}$ in (S16) required by the GMC should be multiplied to the result for $B$. Thus, we complete the proof.

### A.2.3 GMC transition kernel for $\beta_j$ based on $\pi_n^K(\beta_j \mid \cdot)$ under the guidance of $\tilde{\pi}_n^K(\beta_j \mid \cdot)$

First, we focus on the simple case where $d = 1$, i.e., the single index model. The detailed Gibbs sampling method for the multiple index model under the orthonormal constraints are given later.

In the single index model senario, $\pi_n^K(\beta_j \mid \cdot)$ and $\tilde{\pi}_n^K(\beta_j \mid \cdot)$ both degenerate to distributions on the hypersphere $\mathbb{S}^{p-1}$ embedded in $\mathbb{R}^p$. Within this subsection, we denote $\pi_n^K(\beta_j \mid \cdot)$ by $\pi(\beta)$ and $\tilde{\pi}_n^K(\beta_j \mid \cdot)$ by $\tilde{\pi}(\beta)$ for simple notation. Although drawing samples from a distribution on a general Riemannian manifold is often a challenging task (check Liu and Zhu (2022) for a comprehensive review), Byrne and Girolami (2013) have showed that for cases where the Riemannian manifold under consideration enjoys a relatively simple geometry such as simplex or hypersphere, an HMC-based algorithm called geodesic Monte Carlo (GMC) could achieve efficient sampling.

Given an external momentum vector $v$ at location $\beta$ for the target distribution $\pi(\beta)$, the Hamiltonian of the physical system with $\pi(\beta)$ as the potential field is defined as

$$
H(\beta, v) = -\log \pi(\beta) + \frac{1}{2} v^\intercal v. \tag{S13}
$$

The Hamiltionian dynamics of GMC aims to simulate the evolution of $\beta$ along the manifold where $\beta$ is embedded with respect to a random external momentum vector $v$ driven by the Hamiltionian equations. In practice, such an operation can be achieved by a modified version of the leapfrog integrator in HMC. To be concrete, starting from an initial position $\beta(0)$ with initial momentum sampled from a Gaussian distribution, i.e.,

$$
v(0) \sim \mathcal{N}(0, I - \beta(0)\beta(0)^\intercal),
$$

the leapfrog updates $(\beta, v)$ iteratively as following. First, $v(0)$ is updated for a period of $\frac{t}{2}$ and projected to

$$
v(t/2) = (I - \beta(0)\beta(0)^\intercal) \left( v(0) + \frac{t}{2} \nabla_\beta \log \pi(\beta)|_{\beta=\beta(0)} \right), \tag{S14}
$$

where the matrix $(I - \beta(0)\beta(0)^\intercal)$ projects the $p$-dimensional gradient onto the tangent space of the unit sphere at $\beta(0)$, ensuring that the direction of the momentum is tangent to the sphere. Next, we further update $\beta$ and $v$ according to the geodesic flow, which is a rotation along the

great circle on the sphere determined by $v(t/2)$,

$$\beta(t) = \beta(0)\cos(\alpha t) + (vt/\alpha)\sin(\alpha t),$$
$$v(t) = vt\cos(\alpha t) - \alpha\beta(0)\sin(\alpha t), \tag{S15}$$

where $\alpha = ||v||_2$ is the angular velocity. Then, (S14) is applied again from $v(t/2)$ for a period of $\frac{t}{2}$, to get $v(t)$. Applying updates defined in (S14), (S15) and (S14) recursively for $L$ times, we finally obtain the proposed move $(\beta(Lt), v(Lt))$ based on Hamiltonian dynamics of GMC. Note that unlike the classic HMC that may propose move outside the hypersphere, the GMC integrator always keeps $\beta$ staying in the hypersphere and the velocity tangent to the hypersphere, while maintains the time-reversibility and volume preservation property.

However, direct evaluation of the gradient $\nabla \log \pi(\beta)$ is computationally expensive, due to the presence of the tricky term $h(\beta, \Psi_K)$ in (22). To alleviate the computation burden, we suggest simulating the Hamilton's dynamic according to the modified Hamiltonian according to $\tilde{\pi}(\beta)$ instead:

$$\tilde{H}(\beta, v) = -\log \tilde{\pi}(\beta) + \frac{1}{2}v^{\mathsf{T}}v,$$

which is constructed by substituting $\pi(\beta)$ in (S13) by $\tilde{\pi}(\beta)$. Compared to the original Hamiltonian $H(\beta, v)$, the modified Hamiltonian $\tilde{H}(\beta, v)$ leads to an alternative Hamilton's dynamic that is much easier to compute. Let $(\beta^*, v^*) = (\beta(Lt), v(Lt))$ be the final output from the Hamilton's dynamic of GMC based on $\tilde{H}(\beta, v)$ after $L$ leapfrog moves starting from an initial status $(\beta(0), v(0))$. We calculate the Metropolis-Hastings ratio for accepting the GMC proposal $(\beta^*, v^*)$ according to the exact Hamiltonian $H(\beta, v)$ as:

$$r_{\text{GMC}}(\beta^*) = \min\left\{1, \exp\left[H(\beta_0, v_0) - H(\beta^*, v^*)\right]\right\}, \tag{S16}$$

and accept $\beta^*$ with probability $r(\beta^*)$. Algorithm S1 summarizes the whole transition kernel of the GMC move on the unit sphere.

One of the most important ingredients of this algorithm is the momentum direction given by

$$\nabla_{\beta_j} \log \tilde{\pi}(\beta_j) = \nabla_{\beta_j} \log \tilde{\pi}_K(\beta_j \mid \cdot) = -\tilde{M}_j^{-1}(\beta_j - \tilde{\beta}_j). \tag{S17}$$

### A.2.4 Gibbs sampling with orthogonality constraints for multiple index models.

For the model where $d > 1$, the $d$ columns of the matrix $B = (\beta_1, \cdots, \beta_d)$ are updated sequentially within a Gibbs sampling framework. Without loss of generality, we detail the sampling procedure for a single column, $\beta_1$, conditional on the remaining columns, denoted by $B_{[-1]} = (\beta_2, \cdots, \beta_d)$. The full conditional distribution for $\beta_1$ is given by:

$$\pi_{\beta_1}(\beta \mid \cdot) \propto \mathcal{N}(\beta \mid \mu, \Sigma)\mathbb{1}(\|\beta_1\|_{L_2} = 1, \beta_1 \perp B_{[-1]})$$

where $\mathbb{1}(\cdot)$ is the indicator function, enforcing that $\beta_1$ lies on the unit sphere and is orthogonal to the subspace spanned by the other columns.

To efficiently sample from this constrained space, we employ a change of variables. The condition $\beta_1 \perp B_{[-1]}$ restricts $\beta_1$ to the null space of $B_{[-1]}^{\mathsf{T}}$, which is a subspace of dimension

**Algorithm S1** GMC transition kernel for $\beta$ based on $\pi(\beta)$ under the guidance of $\tilde{\pi}(\beta)$
***
 1: **Hyperparameters:** step size $t$, number of steps $L$.
 2: **Input:** Starting point $\beta_0$, target distribution $\pi(\beta)$, auxiliary distribution $\tilde{\pi}(\beta)$.
 3: Sample $v \sim \mathcal{N}(0, I_p - \beta_0\beta_0^\mathsf{T})$;
 4: $h_0 \leftarrow \log\pi(\beta_0) - \frac{1}{2}v^\mathsf{T}v$;
 5: $\beta \leftarrow \beta_0$.
 6: **for** $\tau = 1, 2, \cdots, L$ **do**
 7: $\quad v \leftarrow (I - \beta\beta^\mathsf{T})\left(v + \frac{t}{2}\nabla_\beta\log\tilde{\pi}(\beta)\right)$; $\alpha \leftarrow \|v\|_2$;
 8: $\quad \beta \leftarrow \beta\cos(t\alpha) + (v/\alpha)\sin(t\alpha)$; $v \leftarrow v\cos(t\alpha) - \alpha\beta\sin(t\alpha)$;
 9: $\quad v \leftarrow (I - \beta\beta^\mathsf{T})\left(v + \frac{t}{2}\nabla_\beta\log\tilde{\pi}(\beta)\right)$;
10: **end for**
11: $h \leftarrow \log\pi(\beta) - \frac{1}{2}v^\mathsf{T}v$;
12: Sample $u \sim \mathrm{Unif}(0, 1)$;
13: **if** $u < \exp(h_0 - h)$ **then**
14: $\quad$ **Return** $\beta$;
15: **else**
16: $\quad$ **Return** $\beta_0$.
17: **end if**
***

$p - d + 1$. We can construct an orthogonal matrix $Q \in \mathbb{R}^{p\times p}$, for instance via the Gram-Schmidt procedure, such that its last $p-d+1$ columns form an orthonormal basis for this subspace $B_{[-1]}^\mathsf{T}$.

This allows us to reparameterize any $\beta_1$ satisfying the orthogonality constraint as $\beta_1 = Q\tilde{\gamma}$, where the first $d-1$ elements of the vector $\tilde{\gamma} \in \mathbb{R}^p$ are zero. Specifically, $\tilde{\gamma}$ takes the form:

$$\tilde{\gamma} = \begin{pmatrix} 0_{d-1} \\ \gamma \end{pmatrix},$$

where $\gamma \in \mathbb{R}^{p-d+1}$. The unit norm constraint $\|\beta_1\|_{L_2} = 1$ transforms to $\|\gamma\|_{L_2} = 1$, as $\|Q\tilde{\gamma}\|_{L_2} = \|\tilde{\gamma}\|_{L_2} = \|\gamma\|_{L_2}$. This reduces the problem to sampling $\gamma$ from the unit sphere $\mathcal{S}^{p-d}$. Once a sample for $\gamma$ is obtained, it is transformed back to the original parameter space via $\beta_1 = Q_{[:,d:p]}\gamma$, where $Q_{[:,d:p]}$ is the submatrix formed by the last $p-d+1$ columns of $Q$.

Since the Jacobian of this orthogonal transformation is $|\det(Q)| = 1$, the target probability density for $\tilde{\gamma}$ is derived by substituting $\beta_1 = Q\tilde{\gamma}$ into the original density:

$$\pi(\tilde{\gamma} \mid \cdot) \propto \mathcal{N}(Q\tilde{\gamma} \mid \mu, \Sigma)\mathbb{I}(\|\tilde{\gamma}\|_{L_2} = 1, \tilde{\gamma}_{1:d-1} = \mathbf{0})$$

This is equivalent to a distribution for $\tilde{\gamma}$ proportional to $\mathcal{N}(\tilde{\gamma} \mid Q^\mathsf{T}\mu, Q^\mathsf{T}\Sigma Q)$, subject to the same constraints. By setting the first $d-1$ components of $\tilde{\gamma}$ to zero, the target distribution for $\gamma$ on the sphere $\mathcal{S}^{p-d}$ becomes a Fisher-Bingham distribution:

$$\pi(\gamma \mid \cdot) \propto \mathcal{N}(\gamma \mid \mu_\gamma, \Sigma_\gamma)\mathbb{I}(\|\gamma\|_{L_2} = 1)$$

where the parameters $\mu_\gamma$ and $\Sigma_\gamma$ are the relevant sub-blocks of the transformed parameters:

- Mean vector: $\mu_\gamma = [Q^\mathsf{T}\mu]_{[d:p]}$

- Covariance matrix: $\Sigma_\gamma = [Q^{\mathsf{T}} \Sigma Q]_{[d:p,d:p]}$

For the Geodesic Monte Carlo (GMC) algorithm, the score function is readily computed. The log of the target density (ignoring constants) is proportional to $-\frac{1}{2}(\gamma - \mu_\gamma)^{\mathsf{T}} \Sigma_\gamma^{-1}(\gamma - \mu_\gamma)$. The score function is therefore:

$$\nabla_\gamma \log \pi(\gamma) = -\Sigma_\gamma^{-1}(\gamma - \mu_\gamma).$$

The framework requires a minor modification if a Laplace prior is placed on the columns of $B$:

$$\pi_0(B) \propto \exp\left(-\lambda \sum_{i=1}^{d} \|\beta_i\|_{L_1}\right).$$

In this scenario, the full conditional for $\beta_1$ includes the prior term:

$$\pi(\beta_1 \mid \cdot) \propto \exp(-\lambda\|\beta_1\|_{L_1}) \cdot \mathcal{N}(\beta_1 \mid \mu, \Sigma)\mathbb{I}(\|\beta_1\|_{L_2} = 1, \beta_1 \perp B_{[-1]})$$

Applying the same reparameterization $\beta_1 = Q_{[:,d:p]}\gamma$, the Laplace term transforms to $\exp(-\lambda\|Q_{[:,d:p]}\gamma\|_{L_1})$. The resulting target distribution for $\gamma$ is:

$$\pi(\gamma \mid \cdot) \propto \exp(-\lambda\|Q_{[:,d:p]}\gamma\|_{L_1}) \cdot \mathcal{N}(\gamma \mid \mu_\gamma, \Sigma_\gamma)\mathbb{I}(\|\gamma\|_{L_2} = 1).$$

The inclusion of the Laplace prior introduces an additional term to the score function. The total score function becomes the sum of the gradient from the Gaussian part and the gradient from the new prior term. The gradient of the log-Laplace term is as follows:

$$\begin{aligned}
& \nabla_\gamma \log \exp(-\lambda\|Q_{[:,d:p]}\gamma\|_{L_1}) \\
= & -\lambda\nabla_\gamma\|Q_{[:,d:p]}\gamma\|_{L_1} \\
= & -\lambda(Q_{[:,d:p]})^{\mathsf{T}}\text{sign}(Q_{[:,d:p]}\gamma).
\end{aligned}$$

This gradient is defined at all points where the components of $Q_{[:,d:p]}\gamma$ are nonzero. The complete score function for use in the GMC algorithm is then:

$$\nabla_\gamma \log \pi(\gamma) = -\Sigma_\gamma^{-1}(\gamma - \mu_\gamma) - \lambda(Q_{[:,d:p]})^{\mathsf{T}}\text{sign}(Q_{[:,d:p]}\gamma).$$

## A.3 Additional Simulation Results

### A.3.1 Convergence diagnosis

To monitor and compare the convergence process of the three available Bayesian methods, i.e., SPB, BMM and spLGP, we visualize their trace plots in Figure S1. For fair comparison, in each experiment all competing methods started from a same initial point of $B$ randomly chosen in $\mathcal{B}_{p,d}$. Figure S1a demonstrates typical trace plots for elements of parameter $B$ in SIM $\mathcal{M}_1$, which degenerates to a $p$-dimensional vector in this case, as well as the trace plot of projection Frobenius distance $\mathbf{d}_{pF}$ between the sampled SDR subspace and the true SDR subspace $\mathcal{S}_0$, under SPB, BMM and spLGP. Similar results for the other three single index models in Example 1, i.e., $\mathcal{M}_2$, $\mathcal{M}_3$ and $\mathcal{M}_4$, are visualized in Figure S1b, S1c and S1d, respectively. Considering that

parameter $B$ in MIMs, such as $\mathcal{M}_5$-$\mathcal{M}_8$, have too many elements, we do not show element-level trace plots for $B$ any more for the four MIMs in Example 2, with only the trace plots for $\mathbf{d}_{pF}$ demonstrated in Figure S1e. From these trace plots, the proposed SPB sampler shows stable convergence behavior and often attains smaller $\mathbf{d}_{pF}$ values to the true SDR subspace $\mathcal{S}_0$ than the other Bayesian methods in these representative runs. These results provide additional evidence that SPB is competitive with existing Bayesian approaches for SDR.

### A.3.2 Posterior inference

After enough posterior samples were obtained after the burning-in period, statistical inference about the unknown SDR subspace was conducted. To avoid potential correlation between nearby samples in the MCMC procedure, we choose one posterior sample every 50 MCMC iterations. Figure S2a demonstrated the posterior distribution of $B$'s elements in single index model $\mathcal{M}_2$ ($p = 10$ and $n = 200$) by boxplots based on the obtained posterior samples. Clearly, these posterior distributions are highly informative for statistical inference as well as variable selection for $\mathcal{M}_2$. To show the variable selection ability in multiple index models, we visualize the posterior distribution of each element of the $B$ in Figure S2b for $\mathcal{M}_5$ when $\beta_1 = e_1$ and $\beta_2 = e_2$. From the figure, we can see clearly that only the first two coordinates of $X$ are effectively involved in the established model. But it is not necessary for each index to contain only one coordinate.

Although we mainly focus on the central subspace $\mathcal{S}$, or equivalently its basis $B$, in SDR, the quality of the fitted conditional distribution $\hat{F}_{Y|Z}$ plays a critical role in effectively inferring $\mathcal{S}$. A Bayesian approach that provides flexible fitting of $F_{Y|Z}$ with lower model complexity and computational cost can be advantageous in practice. Figure S3 visualizes the fitted conditional distribution $\hat{F}_{Y|Z}$ by SPB, BMM and spLGP versus the true conditional distribution $F_{Y|Z}$ in the four SIMs in Example 1, i.e., $\mathcal{M}_1$-$\mathcal{M}_4$, respectively. The scatter plot of $(B_0^\intercal x_i, y_i)$ is also showed in these figures, with the conditional mean $E[Y \mid Z]$ highlighted in a red curve. From these figures, $\hat{F}_{Y|Z}$ from SPB is often closer to the true $F_{Y|Z}$ than the corresponding estimates from spLGP and BMM in these representative examples. A similar phenomenon holds for the four MIMs in Example 2, i.e., $\mathcal{M}_5$-$\mathcal{M}_8$, as well. This helps explain why SPB can perform well relative to existing Bayesian SDR methods in settings where flexible conditional-density fitting is important.

## Supplementary References

Simon Byrne and Mark Girolami. Geodesic Monte Carlo on Embedded Manifolds. *Scandinavian Journal of Statistics*, 40(4):825–845, 2013.

Chang Liu and Jun Zhu. Chapter 10 - Geometry in Sampling Methods: A Review on Manifold MCMC and Particle-based Variational Inference Methods. In *Handbook of Statistics*, volume 47, pages 239–293. Elsevier, 2022.

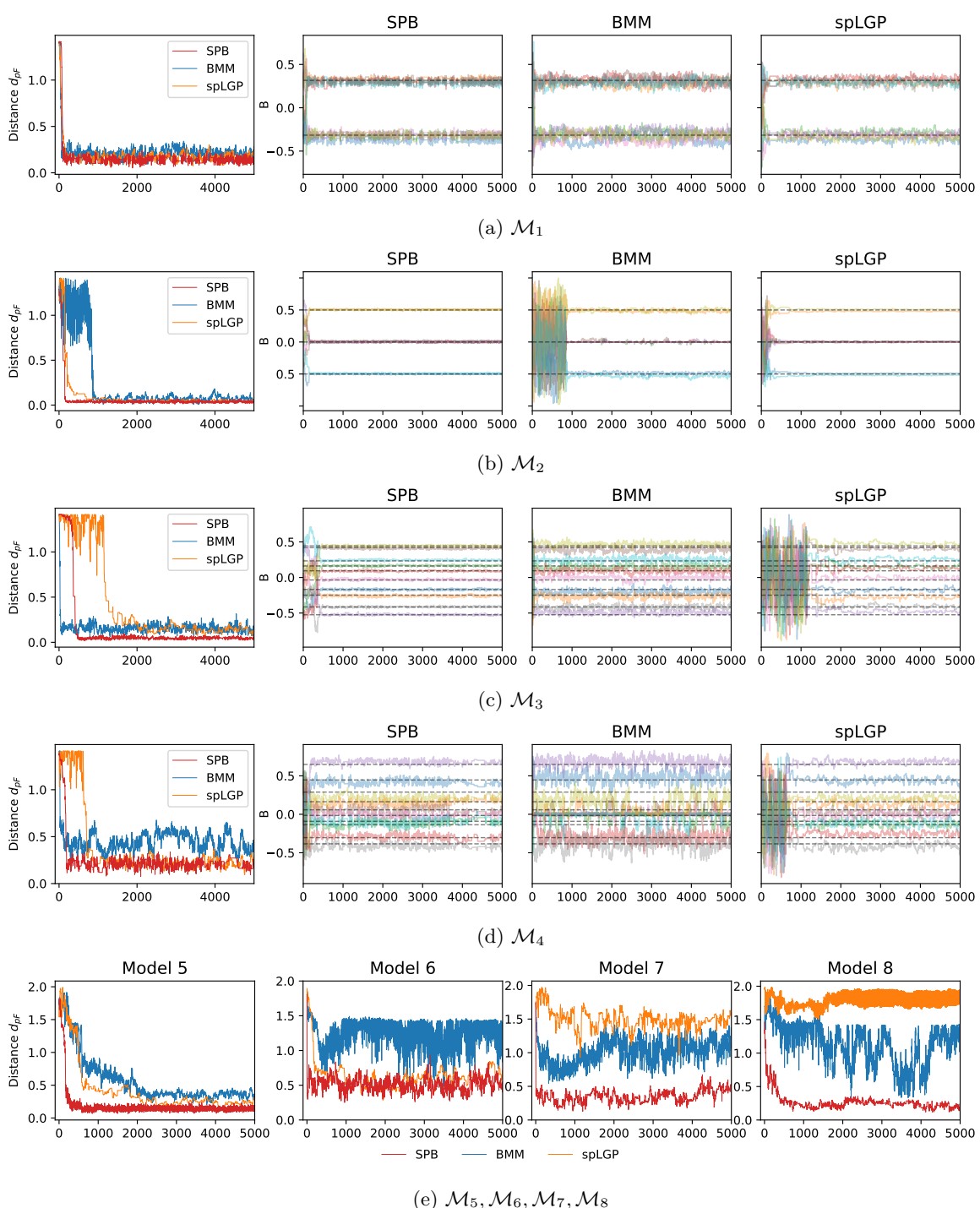

Figure S1: Trace plots and distance between the truth and posterior samples.

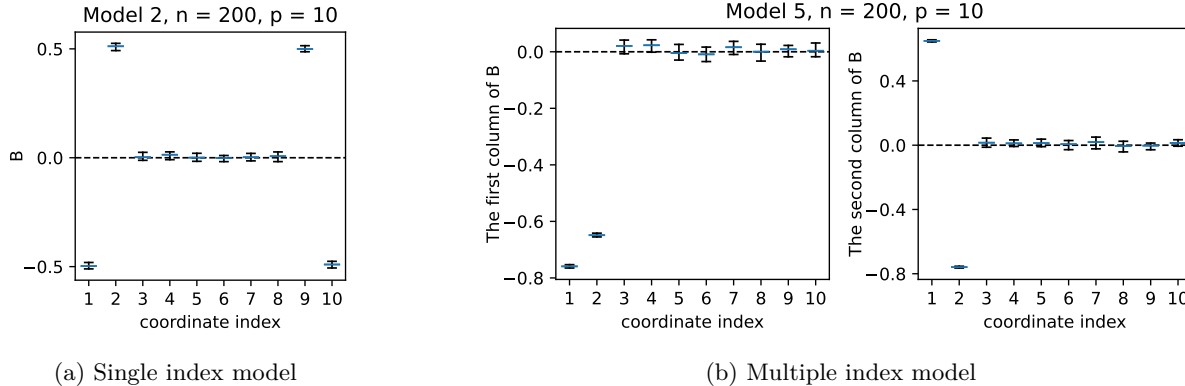

(a) Single index model                    (b) Multiple index model

Figure S2: The quantiles of the posterior samples

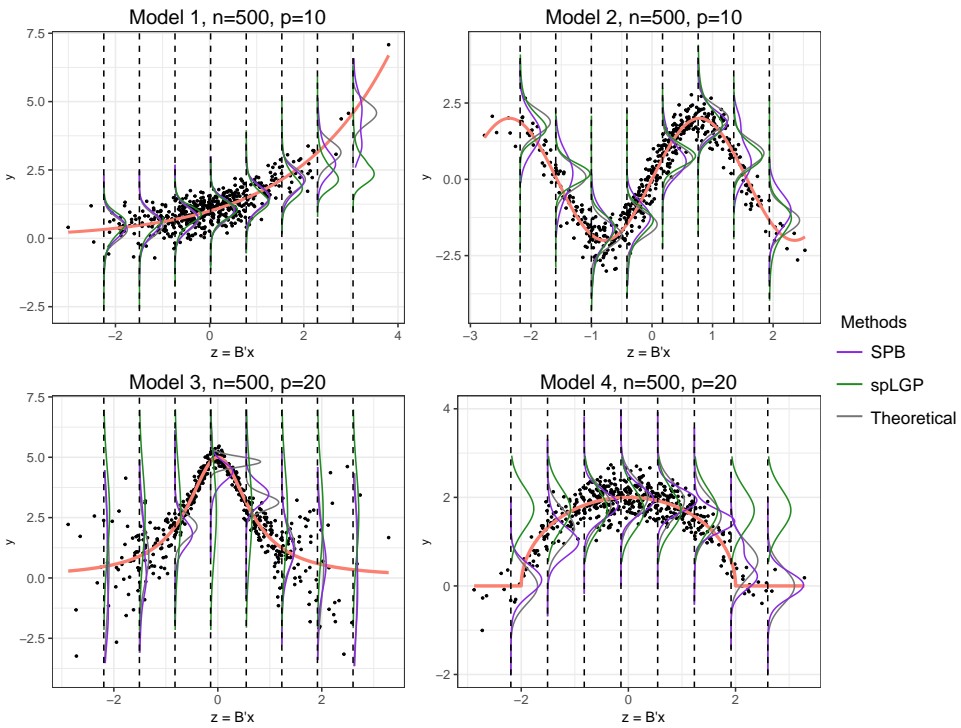

Figure S3: The estimated conditional distributions at some values of the index $Z$