# OpenReview forum: "A Semiparametric Bayesian Method for Sufficient Dimension Reduction"
_SLADS/Section_C — Under review for SLADS_Section_C_

### Review · Reviewer_DAGc · 2026-05-14

**Summary Of Contributions:**

This paper studies sufficient dimension reduction (SDR) and proposes a semiparametric Bayesian (SPB) method for estimating the central subspace. The main idea is to model the joint distribution of $(B^\top X, Y)$ via a Dirichlet process Gaussian mixture, rather than directly modeling the conditional distribution of $Y \mid B^\top X$. The paper also develops a Metropolis-within-Gibbs algorithm for posterior sampling. In addition, the authors establish posterior consistency results that guarantee accurate recovery of the true central subspace under mild conditions.

**Audience:**

Yes

**Broader Impact Concerns:**

I do not have any concerns about the broader impact.

**Claims And Evidence:**

Yes

**Requested Changes:**

*Critical points*:
1. The authors should explain how the proposed method overcomes the limitations of existing Bayesian approaches.
2. The literature review on Bayesian SDR is incomplete. The paper would benefit from a more comprehensive review of this literature.
3. Related to point 4 in *Strengths And Weaknesses*, the authors should clarify the benefit of modeling the joint distribution instead of the conditional distribution.
4. The authors should clarify point 5 in *Strengths And Weaknesses*.
5. For simulation studies, the authors should consider other single index models where the proposal dominates other competitors.
6. Additional comparisons on the real data, or a more substantive scientific exploration of the data, would help demonstrate the practical usefulness of the proposed method.
2. The authors should provide empirical evidence, such as running time comparisons, to support the high computational efficiency of the proposed approach.

*Strengthen-the-work points*:
1. Could the authors provide a more specific definition of $L_d$, which is introduced on page 7?

**Strengths And Weaknesses:**

1. The paper does make contributions to the SDR literature. The main contribution appears to be computational. The authors model the joint distribution of $Y$ and $B^\top X$, whereas conventional Bayesian approaches to SDR model the conditional distribution of $Y \mid B^\top X$.
1. The authors claim that existing methods suffer from either high computational cost or insufficient flexibility in data fitting. However, it is not clearly explained how the proposed method overcomes these limitations.
1. The literature review on Bayesian SDR is incomplete. The authors mention only two works published in 2010 and 2011, while there are a number of more recent developments in Bayesian SDR.
1. Related to the previous point, I am not fully clear about the benefit of modeling the joint distribution instead of the conditional distribution. In equations (5) and (8), the likelihood under the conditional distribution $F_{Y|Z}$ is given. However, in equation (9), the likelihood defined under the joint distribution $F_{Z,Y}$ appears to be equivalent to the likelihood based on $F_{Y|Z}$. What, then, is the key reason for introducing $F_{Z,Y}$? My understanding is that the benefit of using the joint distribution $F_{Z,Y}$ is to avoid the denominator $f_Z(z(B))$ in the likelihood under the conditional distribution $F_{Y|Z}$.
1. I am confused by the transition from $\pi_n$ in equation (23), to $\pi_n^K$ in equation (25), and then to $\tilde\pi_n^K$ in equation (27). If the denominator $h(B,\Psi) = \prod_{i=1}^n f_Z(z_i(B))$ is the computationally challenging part, and if the proposed Algorithm 1 only involves $\tilde\pi_n^K$ in equation (27), which drops the term $h(B,\Psi)$ or $h(B,\Psi_K)$, then why not omit the denominator $f_Z(z(B))$ when defining the likelihood $L_n(B, F_{Z,Y})$ under the joint distribution $F_{Z,Y}$ in equation (9) from the beginning? This issue makes it difficult for me to fully appreciate the methodological contribution of the proposed approach.
1. In the simulation studies, the proposed method does not appear to perform best in some settings. For instance, under Models $M_1$, $M_2$, and $M_4$, the proposed SPB method does not seem to enjoy a clear advantage over the PPR method, especially when $n$ is as large as 1000.
1. The real data analyzed in this paper are relatively old. Moreover, the data analysis is rather brief. Only coefficient estimates are reported.
1. Some notations are not used consistently. For instance, in equation (3), $B$ is defined as the basis of an arbitrary SDR subspace, whereas later it is used as the basis of the central subspace. I suggest reserving $B$ for the basis of the central subspace and using a different notation for the basis of a generic SDR subspace.
1. Some references and definitions should be double-checked. First, in the Introduction Section, it is claimed that Li (1991) proposed the multiple index model (MIM) as $Y = g(\beta_1^\top X, \ldots, \beta_d^\top X, \epsilon)$. However, the MIM was not originally proposed in Li (1991), and the term is more commonly used to refer to model (2) rather than model (1). Second, the notion of an SDR subspace was systematically developed in Cook's monograph {\it Regression Graphics: Ideas for Studying Regressions Through Graphics} (1998, John Wiley \& Sons).

---

> ### Author Response · Authors · 2026-06-20
>
> Thank you for the thoughtful and constructive feedback on our manuscript, "A Semiparametric Bayesian Method for Sufficient Dimension Reduction." We revised the manuscript to clarify the methodological contribution, strengthen the Bayesian SDR literature review, improve the presentation of the model and computation, and temper claims where the empirical results indicate comparable rather than uniformly superior performance. Newly added manuscript text is highlighted in red.
>
> ### Comment DAGc-1: Limitations of existing Bayesian approaches
>
> **Original comment:**
>
> > The authors claim that existing methods suffer from either high computational cost or insufficient flexibility in data fitting. However, it is not clearly explained how the proposed method overcomes these limitations.
> >
> > Requested change: The authors should explain how the proposed method overcomes the limitations of existing Bayesian approaches.
>
> **Response:** Thanks for the insightful comment that we did not provide sufficient explanations previously. While most existing Bayesian methods for SDR directly model the conditional distribution $F_{Y\mid Z}$, our method chooses to model $(Z,Y)$ jointly instead and recovers $F_{Y\mid Z}$ from $F_{Z,Y}$ by conditioning. Conceptually, these two strategies are equivalent because they model the same thing with different parametrizations. The key difference lies on the fact that by modelling $F_{Y\mid Z}$ indirectly via $F_{Z,Y}$, it's straightforward to utilize Dirichlet Process Gaussian Mixture (DPGM) model, a popular tool for non-parametric Bayesian, in our inference procedure. The involvement of DPGM model grants us great computational and theoretical advantages over exiting Bayesian methods for SDR directly modeling $F_{Y\mid Z}$, which require either discretizing a stochastic process over $z$ or using mixture structures with index-dependent weights.
>
> **Changes made:** Section 1 and Section 2.1.
>
> ### Comment DAGc-2: Bayesian SDR literature
>
> **Original comment:**
>
> > The literature review on Bayesian SDR is incomplete. The authors mention only two works published in 2010 and 2011, while there are a number of more recent developments in Bayesian SDR.
> >
> > Requested change: The paper would benefit from a more comprehensive review of this literature.
>
> **Response:** Thanks for pointing out more recent papers along this research line. In the revised paper, we expanded the Bayesian literature discussion. In addition to Tokdar et al. (2010) and Reich et al. (2011), we now mention related Bayesian supervised dimension-reduction work, including Page, Bhattacharya, and Dunson (2013) and Cai, Lin, and Li (2019). The work of Page, Bhattacharya, and Dunson (2013) is connected to ours because it also uses Bayesian nonparametric modeling to learn low-dimensional structure with posterior uncertainty quantification, but it focuses on classification and density learning through affine subspaces in the predictor space rather than estimating the SDR central subspace for a general regression response. Cai, Lin, and Li (2019) is connected to ours through Bayesian supervised dimension reduction and inverse-regression ideas, especially for small datasets, but their method uses the conditional distribution of X given Y to recover the dimension-reduction space, whereas our method uses a forward SDR likelihood, models the joint distribution of (B^T X,Y) with a DPGM prior, and recovers F_{Y|Z} by conditioning.
>
> **Changes made:** Section 1.
>
> ### Comment DAGc-3: Benefit of $F_{Z,Y}$ and Eq. (9)
>
> **Original comment:**
>
> > I am not fully clear about the benefit of modeling the joint distribution instead of the conditional distribution. In equations (5) and (8), the likelihood under the conditional distribution $F_{Y|Z}$ is given. However, in equation (9), the likelihood defined under the joint distribution $F_{Z,Y}$ appears to be equivalent to the likelihood based on $F_{Y|Z}$. What, then, is the key reason for introducing $F_{Z,Y}$? My understanding is that the benefit of using the joint distribution $F_{Z,Y}$ is to avoid the denominator $f_Z(z(B))$ in the likelihood under the conditional distribution $F_{Y|Z}$.
>
> **Response:** We totally agree with your interpretation that Eq. (9) is algebraically the conditional likelihood under a reparameterization. The reason for introducing $F_{Z,Y}$ is not to change the inferential target, but to place a flexible joint-mixture prior on $(Z,Y)$ and isolate the marginal density term for corrected Monte Carlo proposals. The revised manuscript now states this explicitly.
>
> **Changes made:** Section 2.1 and Section 4.1.

---

> > ### Author Response · Authors · 2026-06-20
> >
> > ### Comment DAGc-4: Transition from $\pi_n$ to $\pi_n^K$ to $\tilde{\pi}_n^K$
> >
> > **Original comment:**
> >
> > > I am confused by the transition from $\pi_n$ in equation (23), to $\pi_n^K$ in equation (25), and then to $\tilde{\pi}_n^K$ in equation (27). If the denominator $h(B, \Psi) = \prod_{i=1}^n f_Z(z_i(B))$ is the computationally challenging part, and if the proposed Algorithm 1 only involves $\tilde{\pi}_n^K$ in equation (27), which drops the term $h(B, \Psi)$ or $h(B, \Psi_K)$, then why not omit the denominator $f_Z(z(B))$ when defining the likelihood $L_n(B, F_{Z,Y})$ under the joint distribution $F_{Z,Y}$ in equation (9) from the beginning? This issue makes it difficult for me to fully appreciate the methodological contribution of the proposed approach.
> >
> > **Response:** Sorry that our original texts did not sufficiently distinguish these objects. The revision now states that $\pi_n$ is the exact infinite-DPGM posterior, $\pi_n^K$ is the finite-truncation working target, and $\tilde{\pi}_n^K$ is an auxiliary proposal distribution, not the posterior. The denominator is not omitted from the target; it is removed only in the proposal and restored by the Metropolis-Hastings correction.
> >
> > **Changes made:** Section 4.1 and Algorithm 1 discussion.
> >
> > ### Comment DAGc-5: Simulation dominance
> >
> > **Original comment:**
> >
> > > In the simulation studies, the proposed method does not appear to perform best in some settings. For instance, under Models $M_1$, $M_2$, and $M_4$, the proposed SPB method does not seem to enjoy a clear advantage over the PPR method, especially when $n$ is as large as 1000.
> > >
> > > Requested change: For simulation studies, the authors should consider other single index models where the proposal dominates other competitors.
> >
> > **Response:** We agree that SPB is not uniformly best in every setting. We revised the claims accordingly. The manuscript now states that SPB is frequently among the two best methods and performs competitively, especially in nonlinear, heteroscedastic, or multiple-index settings. We also note that PPR can be highly competitive when the data-generating structure is favorable for projection pursuit. We did not add a new simulation chosen solely to make SPB dominate, because that would be less informative than accurately interpreting the existing broad simulation suite.
> >
> > **Changes made:** Section 5 and the abstract/conclusion language.
> >
> > ### Comment DAGc-6: Real-data analysis depth
> >
> > **Original comment:**
> >
> > > The real data analyzed in this paper are relatively old. Moreover, the data analysis is rather brief. Only coefficient estimates are reported.
> > >
> > > Requested change: Additional comparisons on the real data, or a more substantive scientific exploration of the data, would help demonstrate the practical usefulness of the proposed method.
> >
> > **Response:** Thank you for your suggestions. We expanded the interpretation of the Auto MPG index, the posterior intervals for model year and origin, and the scientific meaning of the two-index concrete-strength fit. We also clarified what the prediction metrics and posterior coefficient summaries show in practice.
> >
> > **Changes made:** Section 6.
> >
> > ### Comment DAGc-7: Computational efficiency evidence
> >
> > **Original comment:**
> >
> > > The authors should provide empirical evidence, such as running time comparisons, to support the high computational efficiency of the proposed approach.
> >
> > **Response:** Thanks for your suggestion. Conceptually, a raw wall-clock comparison table would perfect answer your question. In reality, however, such a table would be difficult to interpret fairly because different methods are implemented in different programming languages: Tokdar's implementation is in C, Reich's implementation is in R, and ours is in Python. In addition, MCMC-based methods require choices of iteration count, burn-in, and thinning; spLGP also requires a logistic-Gaussian-process discretization grid; and frequentist iterative methods depend on tolerances and random restarts. These choices trade speed against accuracy and reliability, so raw wall-clock time could be misleading.
> >
> > Instead, we added a per-iteration complexity analysis for Algorithm 1. Each Gibbs sweep scales as $O(nK(d+1)^2+nKp)$, which is linear in both $n$ and $K$ and avoids a Gaussian-process grid.
> >
> > **Changes made:** Section 4.1.

---

> > > ### Author Response · Authors · 2026-06-20
> > >
> > > ### Comment DAGc-8: Definition of $L_d$
> > >
> > > **Original comment:**
> > >
> > > > Could the authors provide a more specific definition of $L_d$, which is introduced on page 7?
> > >
> > > **Response:** Sorry for this vague definition. The revised dimension-selection subsection now defines $L_d$ as the maximized conditional likelihood under candidate structural dimension $d$, and explains how it is approximated from posterior draws in practice.
> > >
> > > **Changes made:** Section 2.4.
> > >
> > > ### Comment DAGc-9: Notation for $B$
> > >
> > > **Original comment:**
> > >
> > > > Some notations are not used consistently. For instance, in equation (3), $B$ is defined as the basis of an arbitrary SDR subspace, whereas later it is used as the basis of the central subspace. I suggest reserving $B$ for the basis of the central subspace and using a different notation for the basis of a generic SDR subspace.
> > >
> > > **Response:** Sorry for these inconsistent definitions. We revised the opening SDR formulation to use a generic matrix $A$ for an arbitrary SDR subspace and reserve $B$ for an orthonormal basis representing the target central subspace. We also emphasize that inference is for $\mathrm{span}(B)$.
> > >
> > > **Changes made:** Section 1 and Section 2.1.
> > >
> > > ### Comment DAGc-10: References and definitions
> > >
> > > **Original comment:**
> > >
> > > > Some references and definitions should be double-checked. First, in the Introduction Section, it is claimed that Li (1991) proposed the multiple index model (MIM) as $Y = g(\beta_1^\top X, \ldots, \beta_d^\top X, \epsilon)$. However, the MIM was not originally proposed in Li (1991), and the term is more commonly used to refer to model (2) rather than model (1). Second, the notion of an SDR subspace was systematically developed in Cook's monograph *Regression Graphics: Ideas for Studying Regressions Through Graphics* (1998, John Wiley & Sons).
> > >
> > > **Response:** Thank you for your comments. We revised the historical paragraph so that Li (1991) is cited for sliced inverse regression, Cook (1994, 1998) for dimension reduction subspaces and the central subspace, and SIM/MIM are presented as forward-regression representations rather than attributing the full general MIM formulation to Li (1991).
> > >
> > > **Changes made:** Section 1.
> > >
> > > ### Comment DAGc-11: Positive assessment of contribution
> > >
> > > **Original comment:**
> > >
> > > > The paper does make contributions to the SDR literature. The main contribution appears to be computational. The authors model the joint distribution of $Y$ and $B^\top X$, whereas conventional Bayesian approaches to SDR model the conditional distribution of $Y \mid B^\top X$.
> > >
> > > **Response:** We thank the reviewer for this assessment. We revised the contribution statement to make clear that the main contribution is the joint-distribution formulation and the resulting computational strategy for Bayesian SDR.
> > >
> > > **Changes made:** Sections 1 and 7.
> > >
> > > ## Summary of Main Changes for Reviewer DAGc
> > >
> > > - Expanded the Bayesian SDR literature review.
> > > - Clarified the benefit of modeling $(Z,Y)$ jointly while preserving the conditional likelihood target.
> > > - Clarified the roles of $\pi_n$, $\pi_n^K$, and $\tilde{\pi}_n^K$.
> > > - Added per-iteration complexity discussion and explained why raw wall-clock comparisons would be misleading.
> > > - Tempered simulation claims and expanded real-data interpretation.
> > > - Clarified $L_d$, revised notation for $B$, and corrected historical attribution.

---

> > > ### Comment · Reviewer_DAGc · 2026-07-06
> > > **Simulation results**
> > >
> > > 1. (Comment 5) In Section 5.2, the current explanation of the phenomenon that the proposed method is not always dominant is not very informative. Specific explanations of why the proposed method losses dominance should be given, especially provided that this phenomenon is not rare.
> > >
> > > 1. (Comment 6) In Section 6.1, for the posterior intervals, HP and WT seem also miss zero and are possibly significantly contributory to the inference of the value of MPG? Please confirm this.
> > >
> > > 1. (Comment 6) In Section 6.2, Figure 7(a) is non-informative as it is difficult to distinguish the true response from the prediction.

---

> > > > ### Author Response · Authors · 2026-07-07
> > > >
> > > > ### Additional Comment 1: Simulation results
> > > >
> > > > **Original comment:**
> > > >
> > > > > In Section 5.2, the current explanation of the phenomenon that the proposed method is not always dominant is not very informative. Specific explanations of why the proposed method losses dominance should be given, especially provided that this phenomenon is not rare.
> > > >
> > > > **Response:** We agree that the previous explanation was too general and could overstate model-specific interpretations that are not guaranteed. We revised Section 5.2 to give a more defensible explanation. In many settings, the numerical gap between SPB and the best competitor is very small, so the apparent ordering can be affected by Monte Carlo error, finite-replication randomness, and the stochastic nature of the MCMC approximation; in such cases the methods should be viewed as close and comparable. We also clarified that a small fraction of SPB runs may be affected by imperfect MCMC convergence. To make the large simulation study comparable across all models, dimensions, sample sizes, and replications, we used a fixed MCMC length and common default hyperparameters rather than tuning each individual chain. Some difficult chains may therefore not have mixed fully within the fixed budget, and we honestly reported the resulting average performance without excluding those runs. In a real application with one dataset, one would inspect diagnostics, tune computational settings if needed, and increase the number of MCMC iterations to ensure satisfactory mixing.
> > > >
> > > > **Changes made:** Section 5.2.
> > > >
> > > > ### Additional Comment 2: Auto MPG posterior intervals
> > > >
> > > > **Original comment:**
> > > >
> > > > > In Section 6.1, for the posterior intervals, HP and WT seem also miss zero and are possibly significantly contributory to the inference of the value of MPG? Please confirm this.
> > > >
> > > > **Response:** The reviewer is correct. The posterior interval for HP misses zero, and the WT interval is close to zero but still separated from it in the plotted posterior summary. We revised Section 6.1 to state that HP and WT should also be regarded as contributory variables in this fitted single-index model. We also added a caution that their effects are smaller on the standardized index scale than those of MY and OG, and that the sign of coefficients is arbitrary up to the orientation of the one-dimensional SDR subspace.
> > > >
> > > > **Changes made:** Section 6.1.
> > > >
> > > > ### Additional Comment 3: Concrete data Figure 7(a)
> > > >
> > > > **Original comment:**
> > > >
> > > > > In Section 6.2, Figure 7(a) is non-informative as it is difficult to distinguish the true response from the prediction.
> > > >
> > > > **Response:** We agree that the previous presentation did not clearly explain the visual encoding. Since this is a real-data example, panel (a) compares observed responses, not a known true response surface, with the fitted posterior predictive surface. We enlarged panel (a), revised its caption to state that red points are observed CCS values and the colored mesh is the SPB posterior predictive mean, and added text emphasizing that the quantitative agreement is summarized by the reported $R^2$ and cross-validated RMSE.
> > > >
> > > > **Changes made:** Section 6.2 and Figure 7 caption/layout.

---

> > ### Comment · Reviewer_DAGc · 2026-07-06
> > **More explanation of modeling the joint distribution**
> >
> > 1. If I understand correctly, although formulas (5) and (9) are mathematically equivalent, they appear to correspond to two different modeling strategies: approaches based on (5) model the conditional distribution (Y \mid Z), whereas those based on (9) model the joint distribution ((Y, Z)). Is this interpretation correct? If so, the transition from (5) to (9) seems rather direct, as it follows simply from the definition of conditional density. Thus, the additional equations (6)–(8) may not be necessary and, in fact, make this transition somewhat confusing.
> >
> > 2. Although the authors have provided a general discussion in Section 1 on the disadvantages of directly modeling the conditional distribution, it would be helpful to include more technical details, for example in Section 2, to further clarify the statement that “These choices may increase computational cost or constrain the flexibility of the conditional distribution” (page 3, Section 1).

---

> > > ### Author Response · Authors · 2026-07-07
> > >
> > > ### Comment 1: Equations (5) and (9)
> > >
> > > **Original comment:**
> > >
> > > > If I understand correctly, although formulas (5) and (9) are mathematically equivalent, they appear to correspond to two different modeling strategies: approaches based on (5) model the conditional distribution $(Y \mid Z)$, whereas those based on (9) model the joint distribution $((Y, Z))$. Is this interpretation correct? If so, the transition from (5) to (9) seems rather direct, as it follows simply from the definition of conditional density. Thus, the additional equations (6)-(8) may not be necessary and, in fact, make this transition somewhat confusing.
> > >
> > > **Response:** This interpretation is correct. Equations (5) and (9) are algebraically the same conditional likelihood, but they correspond to different prior-modeling strategies: direct conditional modeling of $F_{Y\mid Z}$ versus joint modeling of $F_{Z,Y}$ followed by conditioning. We revised Section 2.1 to make this point explicit and streamlined the derivation. The previous intermediate construction through the joint density of $(X,Y)$ has been removed, and the transition is now presented directly through the conditional-density identity.
> > >
> > > **Changes made:** Section 2.1.
> > >
> > > ### Comment 2: Technical explanation of conditional versus joint modeling
> > >
> > > **Original comment:**
> > >
> > > > Although the authors have provided a general discussion in Section 1 on the disadvantages of directly modeling the conditional distribution, it would be helpful to include more technical details, for example in Section 2, to further clarify the statement that "These choices may increase computational cost or constrain the flexibility of the conditional distribution" (page 3, Section 1).
> > >
> > > **Response:** We added the requested technical details in Section 2.1. The revision now shows that a Gaussian mixture prior on the joint density $f_{Z,Y}$ automatically induces a conditional mixture for $Y\mid Z=z$, with $z$-dependent weights, means, and variances available in closed form. In contrast, a direct nonparametric conditional-density model must specify normalized positive functions of $z$ for the mixture weights, and often functions of $z$ for component parameters, which typically requires stochastic-process discretization, basis expansion, or nonconjugate updates. This explains more concretely why direct conditional modeling can be computationally heavier or less flexible in practice.
> > >
> > > **Changes made:** Section 2.1.

---

### Review · Reviewer_pMcf · 2026-06-06

**Summary Of Contributions:**

This paper proposes a new Bayesian approach for sufficient dimension reduction. The method models the joint distribution $F_{Z,Y}$ directly rather than modeling the conditional distribution $F_{Y|Z}$. The authors propose the model, specify the prior and posterior distributions, and provide the posterior based inference procedures. Order determination is also conducted, and the authors provide some Monte Carlo methods for posterior sampling. Theoretical justifications for consistency are also provided. The authors also conduct some simulation studies and apply the method to two real datasets.

**Audience:**

Yes

**Broader Impact Concerns:**

I would concern how much improvement the proposed method has compared to the existing Bayesian SDR methods, which model $F_{Y|Z}$ directly, either from the theoretical perspective or the computational perspective. I hope this point can be clearly illustrated in the paper.

**Claims And Evidence:**

Yes

**Requested Changes:**

1. It is weird that the authors list PPR, MAVE and semiparametric methods in the first category in the literature review. However, at least, MAVE is estimating the central mean space, but semiparametric methods are used to estimate the central space (or central mean space). Also, Fukumizu and Leng (2014) appear both in the first and third categories. There are a bunch of papers about SDR methods, and I suggest the authors categorize them carefully, briefly indicating why certain methods belong to some category.
2. I would suggest illustrating that the identifiable parameter is $\mathcal{S} = \mathrm{span}(B)$ when talking about the model in Section 2.1. This is the estimand you will need, which should be clearly identifiable as inference is made afterwards, especially when the outer model $F_{Y|Z}$ is fully nonparametric.
3. Based on the paper, I did not well understand why the proposed method is better than modeling the conditional distribution $F_{Y|Z}$ theoretically or computationally. Could the authors give more discussions on the advantages?
4. The infinite sum in (24) is replaced by a truncated version with $K$ mixture components. What is the price to pay when the authors conduct the truncation? Is there any guarantee that the truncation effect is negligible in this case? Out of curiosity, would the consistency result be affected in this case?
5. I am also wondering if there are any methods to choose the tuning parameter $\lambda$ in $\pi_0(B)$.
6. I would also suggest giving some posterior convergence rate results in the theoretical analysis, if possible. Currently there is only consistency. Though the tests can be done based on the empirical distribution of $\{ d (\mathcal{S}^{(t)}, \mathcal{S}) \}_{1\le t \le T}$, we want to estimate the central space as accurately as possible.
7. Minor issue: Please add the section numbers in the supplementary materials. Currently the reference labels in the manuscript are hard to be identified.

**Strengths And Weaknesses:**

Strengths:
1. Provide the flexible framework of semiparametric Bayesian SDR method, where the joint distribution $F_{Z,Y}$ is modeled.
2. Given posterior consistency guarantee in the theoretical analysis.
3. Propose Monte Carlo methods for implementing the posterior inference.
4. Give the procedures for order determination.

Weakness:
1. Lack of convergence rate guarantees for the estimated subspace.
2. Lack of identifiability illustrations on the estimand $\mathcal{S} = \mathrm{span}(B)$.
3. Need clear illustrations about the advantages of the proposed method compared to existing Bayesian SDR methods.

---

> ### Author Response · Authors · 2026-06-20
>
> Thank you for the thoughtful and constructive feedback on our manuscript, "A Semiparametric Bayesian Method for Sufficient Dimension Reduction." We revised the manuscript to clarify the methodological contribution, strengthen the literature review, improve the presentation of the model and computation, and temper claims where the empirical results indicate comparable rather than uniformly superior performance. Newly added manuscript text is highlighted in red.
>
> ### Comment pMcf-1: Literature categorization
>
> **Original comment:**
>
> > It is weird that the authors list PPR, MAVE and semiparametric methods in the first category in the literature review. However, at least, MAVE is estimating the central mean space, but semiparametric methods are used to estimate the central space (or central mean space). Also, Fukumizu and Leng (2014) appear both in the first and third categories. There are a bunch of papers about SDR methods, and I suggest the authors categorize them carefully, briefly indicating why certain methods belong to some category.
>
> **Response:** We agree that the original categorization was too coarse. We reorganized the frequentist SDR review by the information used: forward-regression and conditional-mean approaches, inverse-regression approaches, sparse/high-dimensional extensions, and gradient-based methods. We also corrected the placement of PPR and removed the duplicate placement of Fukumizu and Leng (2014).
>
> **Changes made:** Section 1, literature-review paragraphs.
>
> ### Comment pMcf-2: Identifiable estimand
>
> **Original comment:**
>
> > I would suggest illustrating that the identifiable parameter is $\mathcal{S} = \text{span}(B)$ when talking about the model in Section 2.1. This is the estimand you will need, which should be clearly identifiable as inference is made afterwards, especially when the outer model $F_{Y|Z}$ is fully nonparametric.
>
> **Response:** Sorry for the confusions. We now explicitly state that the identifiable estimand is the central subspace $\mathcal{S}=\mathrm{span}(B)$, not a particular basis matrix $B$. We explain that $B$ and $BQ$ represent the same Grassmann point for any orthogonal matrix $Q$.
>
> **Changes made:** Section 1 and Section 2.1.
>
> ### Comment pMcf-3: Advantage over modeling $F_{Y\mid Z}$ directly
>
> **Original comment:**
>
> > Based on the paper, I did not well understand why the proposed method is better than modeling the conditional distribution $F_{Y|Z}$ theoretically or computationally. Could the authors give more discussions on the advantages?
>
> **Response:** Sorry that this key point was well explained previously. The revised texts now explicitly states that the joint model $F_{Z,Y}$ offers a flexible low-dimensional DPGM representation for the joint distribution of $(Z,Y)$, from which the conditional distribution $F_{Y\mid Z}$ can be conveniently recovered and remains the inferential target. This representation also creates tractable latent allocation updates after finite truncation. Existing Bayesian methods for SDR that directly model the conditional distribution $F_{Y\mid Z}$, however, do not enjoy these computational and theoreticl advantages, because they require either discretizing a stochastic process over $z$ or using mixture structures with index-dependent weights.
>
> **Changes made:** Section 1, Section 2.1, and Section 4.1.
>
> ### Comment pMcf-4: Truncation of the infinite mixture
>
> **Original comment:**
>
> > The infinite sum in (24) is replaced by a truncated version with $K$ mixture components. What is the price to pay when the authors conduct the truncation? Is there any guarantee that the truncation effect is negligible in this case? Out of curiosity, would the consistency result be affected in this case?
>
> **Response:** We added a discussion of the truncation. The revised manuscript clarifies that the theoretical consistency result is stated for the non-truncated DPGM model, while the finite $K$ truncation is a computational approximation. We also explain that the residual stick-breaking mass beyond $K$ decreases geometrically, so a sufficiently large $K$ gives a small finite-sample approximation error. To check this empirically, we added a separate sensitivity study for $K$ using two single-index models with different levels of joint-density complexity. We compared $K=2,3,5,10,20,30$ over 30 replications and reported both median subspace error and the mean number of occupied mixture components. The results show that estimation stabilizes once $K$ is large enough and that the number of occupied components remains well below the default $K=30$, supporting the claim that the truncation is not binding in the reported experiments.
>
> **Changes made:** Section 4.1 and Section 5.6.

---

> > ### Author Response · Authors · 2026-06-20
> >
> > ### Comment pMcf-5: Choosing $\lambda$
> >
> > **Original comment:**
> >
> > > I am also wondering if there are any methods to choose the tuning parameter $\lambda$ in $\pi_0(B)$.
> >
> > **Response:** The revised manuscript recommends $\lambda=0$ when sparsity is not expected and selecting $\lambda$ by validation or cross-validation based on posterior predictive performance when sparse SDR directions are plausible. We also clarify that $\lambda=20$ in the shrinkage illustration is illustrative rather than a universal default. To make the selection rule concrete, we added a new subsection demonstrating 5-fold cross-validation for $\lambda$ in a sparse single-index model. For each candidate value $\lambda=0,5,10,15,20,25,30$, we compute the cross-validated negative log predictive density (CV-NLPD) from the posterior predictive density, together with CV-RMSE and the projection Frobenius distance for reference. In the reported experiment, CV-NLPD is minimized at $\lambda=5$, with the one-standard-error rule selecting $\lambda=10$; both values are in the range that gives accurate subspace recovery. This supports a fully data-driven predictive criterion for choosing $\lambda$.
> >
> > **Changes made:** Section 4.4, Section 5.4, and Section 5.5.
> >
> > ### Comment pMcf-6: Posterior convergence rates
> >
> > **Original comment:**
> >
> > > I would also suggest giving some posterior convergence rate results in the theoretical analysis, if possible. Currently there is only consistency. Though the tests can be done based on the empirical distribution of $d(\mathcal{S}^{(t)}, \mathcal{S})_{1 \le t \le T}$, we want to estimate the central space as accurately as possible.
> >
> > **Response:** We attempted to strengthen the rate discussion without overstating what the current proof establishes. The revised manuscript adds a formal conditional contraction-rate statement: if the usual prior small-ball and exponentially consistent test conditions hold at rate $\varepsilon_n$, then the posterior contracts around the true central subspace at rate $\varepsilon_n$. We also state clearly that deriving an explicit optimal rate for the DPGM-induced semiparametric SDR model remains future work.
> >
> > **Changes made:** Section 3 and Section 7.
> >
> > ### Comment pMcf-7: Supplementary section numbers
> >
> > **Original comment:**
> >
> > > Minor issue: Please add the section numbers in the supplementary materials. Currently the reference labels in the manuscript are hard to be identified.
> >
> > **Response:** Sorry for the confusing section numbers in the supplementary materials. We converted the supplementary headings to numbered sections and replaced old supplement references with explicit Supplementary Section references.
> >
> > **Changes made:** Supplementary Material and relevant cross-references in Sections 3-5.
> >
> > ### Comment pMcf-8: Broader impact concern on improvement over existing Bayesian SDR
> >
> > **Original comment:**
> >
> > > I would concern how much improvement the proposed method has compared to the existing Bayesian SDR methods, which model $F_{Y|Z}$ directly, either from the theoretical perspective or the computational perspective. I hope this point can be clearly illustrated in the paper.
> >
> > **Response:** The revised manuscript clarifies that the contribution is a joint-distribution reparameterization that preserves the conditional SDR likelihood while improving modeling flexibility and posterior computation. We also added the per-iteration complexity analysis of Algorithm 1.
> >
> > **Changes made:** Sections 1, 2.1, 4.1, and 7.
> >
> > ## Summary of Main Changes for Reviewer pMcf
> >
> > - Reorganized the SDR literature review and corrected method categorization.
> > - Added explicit identifiability discussion for $\mathcal{S}=\mathrm{span}(B)$.
> > - Clarified that the joint formulation preserves the conditional likelihood while improving modeling and computation.
> > - Added truncation discussion and a dedicated $K$ sensitivity study using occupied-component counts and subspace-error summaries.
> > - Added predictive-performance guidance and a 5-fold cross-validation illustration for choosing $\lambda$.
> > - Added a conditional posterior contraction-rate statement and a limitation on explicit optimal rates.
> > - Numbered the supplementary material and updated supplement cross-references.

---

> > ### Comment · Reviewer_pMcf · 2026-06-21
> > **Literature review issues**
> >
> > I am still concerned about the literature review. At least, Ma and Zhu (2012) does not simply model $E(Y|X)$, but considers the full distribution $Y|X$. Also, OPG in the last category is designed to estimate the central mean space, like MAVE, while its variants can be used to estimate the central space. I suggest the authors cite and classify the existing methods properly before publication.

---

> > > ### Author Response · Authors · 2026-06-22
> > >
> > > Thank you for pointing out these inaccuracies. We revised the literature review again and highlighted the new changes in blue.
> > >
> > > Specifically, we no longer classify Ma and Zhu (2012) as a conditional-mean method; instead, we place it in a semiparametric/distributional category because it uses estimating equations derived from the full conditional distribution $Y\mid X$ and can target the central subspace. We also moved OPG to the forward-regression/conditional-mean category together with MAVE, because ordinary OPG is designed for the central mean subspace. We then separated gradient-based variants that use conditional-distribution or classification information and can estimate dimension-reduction spaces beyond the central mean subspace.

---

### Review · Reviewer_JJ65 · 2026-06-25

**Summary Of Contributions:**

This paper proposes a new semiparametric Bayesian (SPB) approach for sufficient dimension reduction (SDR). Unlike existing Bayesian SDR methods that directly model the conditional distribution $Y|B^T X$, the proposed method reparameterizes the problem by modeling the joint distribution of $(B^T X,Y)$  using a Dirichlet Process Gaussian Mixture (DPGM). The authors develop the corresponding Bayesian framework, including prior specification, posterior inference, dimension determination via BIC, and a Metropolis-within-Gibbs sampling algorithm combining Gibbs updates and Geodesic Monte Carlo. Theoretical analysis establishes posterior consistency for recovery of the central subspace. Simulation studies and real-data applications are provided to demonstrate the empirical performance of the proposed method.

**Audience:**

Yes

**Broader Impact Concerns:**

no concerns.

**Claims And Evidence:**

Yes

**Requested Changes:**

1. Provide more insights and discussions, if not a full proof of consistency, for the estimation of $d$
2. Provide complexity analysis, and/or other justifications of using the proposed method with larger $p$
3. Add sensitivity analysis about choice of hyperparameters.

**Strengths And Weaknesses:**

Strengths: The Bayesian formulation seems new and more or less complete. The proposed framework is conceptually appealing and differs from existing Bayesian SDR approaches such as Tokdar et al. (2010) and Reich et al. (2011). The use of Dirichlet Process Gaussian Mixtures provides substantial flexibility for capturing complex nonlinear relationships without requiring restrictive parametric assumptions. The posterior consistency for the estimated SDR subspace is established under suitable regularity conditions. Furthermore, the manuscript develops procedures for point estimation, credible regions, prediction, and dimension determination, making the proposed approach a relatively complete Bayesian SDR framework.

Weaknesses: 1. The proposed BIC-based dimension determination procedure appears reasonable, but no consistency result or theoretical analysis is provided for the estimator of d. The paper relies primarily on simulation evidence. 2. Computational scalability is unclear. The simulation studies only consider dimensions up to
. It remains unclear how the proposed approach scales to modern high-dimensional SDR settings. 3. The method involves several hyperparameters, including DPGM truncation level K, concentration parameter α, and prior specifications. The manuscript provides default settings but does not systematically investigate robustness to these choices.

---

> ### Author Response · Authors · 2026-06-29
>
> Thank you for the thoughtful and constructive feedback on our manuscript. We revised the manuscript to clarify the dimension-selection criterion, sharpen the computational scalability discussion, and add more systematic hyperparameter sensitivity analyses. Newly added manuscript text is highlighted in red.
>
> ### Comment JJ65-1: Dimension determination by BIC
>
> **Original comment:**
>
> > The proposed BIC-based dimension determination procedure appears reasonable, but no consistency result or theoretical analysis is provided for the estimator of d. The paper relies primarily on simulation evidence.
> >
> > Requested change: Provide more insights and discussions, if not a full proof of consistency, for the estimation of $d$.
>
> **Response:** Thanks for pointing out this gap in the original presentation. We agree that the simulation evidence alone did not provide enough insight into why the BIC rule should work. In the revised manuscript, we added a discussion of the usual BIC separation logic. If $d<d_0$, the candidate model misses part of the true central subspace, and under the uniqueness condition used in the consistency theorem its best conditional likelihood is separated from the true model by a positive Kullback--Leibler gap; this gives an $O(n)$ likelihood loss, which dominates the $O(\log n)$ BIC penalty. If $d>d_0$, the extra directions are redundant for $Y\mid B^\top X$, so the likelihood improvement is expected to be only $O_p(1)$ under a regular local likelihood expansion, while the penalty still grows as $O(\log n)$. We also state explicitly that a complete model-selection consistency theorem with the DPGM nuisance prior and the Monte Carlo approximation of $L_d$ remains future work.
>
> **Changes made:** Section 2.4 and Section 7.
>
> ### Comment JJ65-2: Computational scalability and larger $p$
>
> **Original comment:**
>
> > Computational scalability is unclear. The simulation studies only consider dimensions up to . It remains unclear how the proposed approach scales to modern high-dimensional SDR settings.
> >
> > Requested change: Provide complexity analysis, and/or other justifications of using the proposed method with larger $p$.
>
> **Response:** We agree that the scalability of the current implementation needed to be stated more clearly. The revised manuscript now separates the mixture-evaluation cost from the dense linear algebra used to update $B$. The mixture-evaluation part of each Gibbs sweep scales as $O\{nK(d+1)^2+nKp\}$, which is linear in $n$ and $K$ for fixed $d$. We also added an important qualification: the Fisher--Bingham/GMC proposal for updating $B$ involves dense linear algebra of order $O\{d(np^2+p^3)\}$ per sweep in our current implementation. Thus, the present implementation is appropriate for moderate-dimensional problems such as those considered in the simulations ($p\le 50$), while truly high-dimensional SDR would require sparse or low-rank linear algebra and/or preliminary screening. We also note that the Laplace shrinkage prior can reduce the effective dimension when sparse SDR directions are plausible.
>
> **Changes made:** Section 4.2, Section 4.3, and Section 7.

---

> > ### Author Response · Authors · 2026-06-29
> >
> > ### Comment JJ65-3: Sensitivity to hyperparameters
> >
> > **Original comment:**
> >
> > > The method involves several hyperparameters, including DPGM truncation level K, concentration parameter α, and prior specifications. The manuscript provides default settings but does not systematically investigate robustness to these choices.
> > >
> > > Requested change: Add sensitivity analysis about choice of hyperparameters.
> >
> > **Response:** We agree. The earlier revision already separated the choice of the Laplace shrinkage parameter $\lambda$ from the sensitivity analysis for the truncation level $K$. To further address your concern, we added a new sensitivity study for the DPGM concentration parameter $\alpha$. Since $\alpha$ is not fixed in our default implementation but assigned a Gamma hyperprior and updated within the MCMC sampler, we varied the second-level hyperparameters $(\eta_1,\eta_2)$ over $(0.5,0.5)$, $(1,1)$, $(2,1)$, and $(5,1)$. For two representative dense models, we report the projection Frobenius distance, validation RMSE, the number of occupied mixture components, and the posterior mean of $\alpha$. The results show that subspace estimation and prediction are stable across these hyperprior choices. We also added text explaining that, after standardization, the Normal--Inverse-Wishart defaults are weakly informative, and that the main sensitivity analyses focus on $K$, $\alpha$, and $\lambda$ because they directly control mixture complexity and structural sparsity.
> >
> > **Changes made:** Section 4.3, Section 5.5, Section 5.6, Section 5.7, and the new Table on $\alpha$ sensitivity.
> >
> > ## Summary of Main Changes for Reviewer JJ65
> >
> > - Added theoretical discussion of why the BIC rule is expected to recover the true structural dimension under standard separation and regularity conditions.
> > - Expanded the computational complexity discussion, including the dense $B$-update bottleneck for large $p$.
> > - Added a new $\alpha$ hyperprior sensitivity experiment and table.
> > - Clarified that $K$, $\alpha$, and $\lambda$ are the primary hyperparameters affecting mixture complexity and sparsity, while the NIW defaults are weakly informative after standardization.

---

> ### Comment · Reviewer_JJ65 · 2026-07-08
> **The authors have addressed my three comments successfully.**
>
> No further comment to improve

---

### Comment · Action_Editor_UqjQ · 2026-07-18
**Recommendation**

Since all reviewers are satisfied with the revision, I recommend accepting the paper as is.